# On the Limits of Test-Time Compute:
# Sequential Reward Filtering for Better Inference

**Yue Yu** [1]   **Qiwei Di** [2]   **Quanquan Gu** [2]   **Dongruo Zhou** [3]

## Abstract

Test-time compute (TTC) has become an increasingly prominent paradigm for enhancing large language models (LLMs). Despite the empirical success of methods such as best-of-$n$ (BoN) sampling and sequential revision, their fundamental limits remain unclear. We address this gap by analyzing a mixture-of-reference policy model and proving that standard BoN is inherently suboptimal. To move closer to the optimal frontier, we study reward-filtered sequential inference, a simple procedure that selectively incorporates only high-reward generations into the context. This mechanism concentrates computation on superior policy candidates and suppresses inferior ones. On the theoretical side, we show that reward-filtered sequential inference yields strictly stronger guarantees than standard TTC paradigms. On the empirical side, we evaluate such an inference strategy across diverse benchmarks and observe consistent improvements over widely used approaches, demonstrating the practical effectiveness of our framework.

## 1 Introduction

How to effectively utilize large language models (LLMs) for solving new tasks has become a central research question. Among the many approaches, Test-Time Compute (TTC) has recently attracted significant attention. The key idea of TTC is to allocate additional computation during inference to improve task performance. Unlike post-training approaches such as fine-tuning or reinforcement learning, TTC

[1]Department of Statistics, Indiana University Bloomington, Bloomington, IN 47405, USA [2]Department of Computer Science, University of California, Los Angeles, Los Angeles, CA 90095, USA [3]Department of Computer Science, Indiana University Bloomington, Bloomington, IN 47408, USA. Correspondence to: Quanquan Gu <qgu@cs.ucla.edu>, Dongruo Zhou <dz13@iu.edu>.

*Proceedings of the $43^{rd}$ International Conference on Machine Learning*, Seoul, South Korea. PMLR 306, 2026. Copyright 2026 by the author(s).

requires no additional training of the base model. As a result, inference-time alignment methods provide a lightweight yet powerful alternative that greatly simplifies deployment. Well-known TTC methods include Best-of-N (BoN) sampling, chain-of-thought (CoT) reasoning, and their many variants (Stiennon et al., 2020; Nakano et al., 2021; Wei et al., 2022; Wang et al., 2022; Zhou et al., 2022; Yao et al., 2023b;a; Chen et al., 2022; Shinn et al., 2023).

In this work, we specifically focus on a family of TTC methods that leverage an external reward model, which can assign a score to each generated answer. Perhaps the most basic and widely studied approach in this family is BoN, a representative strategy of *parallel TTC*. Given a problem $x$, the LLM generates $n$ answers $a_1, \ldots, a_n$, and then selects the one with the highest reward score. BoN is highly intuitive, easy to implement, and has been empirically shown to substantially outperform vanilla decoding across a wide range of tasks. Naturally, this raises a fundamental question:

*Is Best-of-N the best one can hope for under a fixed test-time budget?*

Recent work (Snell et al., 2024) suggests that the answer may be negative. Unlike BoN, which allocates its test-time budget in parallel by generating multiple independent samples, another line of research explores *sequential TTC*. In sequential TTC, generated answers are inserted back into the input window of the LLM, effectively reshaping its conditional distribution as more computation is spent. Empirically, this approach has demonstrated performance gains beyond BoN by gradually biasing the model towards higher-quality outputs. However, despite its practical promise, the theoretical understanding of sequential TTC remains very limited. This motivates our second central question:

*What is the best achievable strategy under sequential TTC?*

In this work, we formalize TTC with reward models as a decision problem over this mixture-of-reference-policies model and use it to address both questions above. Conceptually, our goals are two-fold: (i) characterize the best-achievable regret under a fixed test-time budget for parallel TTC algorithms such as BoN, and (ii) design a sequential

TTC procedure that, using the same reward model, provably attains a strictly better budget–performance trade-off. Our key contributions are summarized as follows:

- **Limits of parallel TTC.** We study the fundamental limit of TTC under a *mixture-of-reference policies* model, where the pretraining data of the LLM is assumed to consist of trajectories generated by multiple underlying policies. Within this setting, we establish a lower bound on the test-time budget required to achieve a near-optimal policy and show that parallel TTC methods, including BoN, fall short of this bound, indicating that despite its popularity, parallel TTC is not optimal under more realistic modeling assumptions.

- **A reward-filtered sequential TTC method.** To go beyond parallel TTC, we study a simple sequential TTC method called *Reward-Filtered Sequential Best-of-n* (RF-SeqBoN), where only high-reward generations are fed back into the LLM's input window. This procedure progressively refines the generation distribution towards that of the optimal policy induced by the reward function. We prove that, under mild assumptions on the reward model, our method strictly improves upon parallel TTC and achieves significantly larger gains on harder tasks.

- **Experiments.** We conduct extensive experiments across diverse benchmarks and backbone LLMs. Empirical results consistently demonstrate that our approach achieves higher *test-time budget efficiency* than existing baselines, confirming that the theoretical advantages of our method translate into substantial practical improvements.

In the rest of our paper, we formalize the sequential sample-and-evaluate framework for TTC with reward models and specify our mixture-of-reference-policies pretraining assumption in Section 3. Section 4 then develops fundamental lower bounds on the test-time budget required by parallel TTC schemes, showing that vanilla BoN is statistically sub-optimal under this model. Building on these insights, Section 5 introduces our *Reward-Filtered Sequential Best-of-$n$ (RF-SeqBoN)* algorithms, together with assumptions on the reward model, and establishes improved sample-complexity guarantees that strictly dominate parallel TTC in appropriate regimes. Section 6 presents empirical evaluations across multiple benchmarks and backbone LLMs, including ablations on key design choices, demonstrating consistent gains in budget efficiency. Finally, Section 7 concludes with a summary of our theoretical and empirical work.

**Notation.** We use lowercase letters to denote scalars, and bold lowercase (resp. uppercase) letters to denote vectors (resp. matrices). For an integer $n \in \mathbb{N}$, let $[n] := \{1, 2, \ldots, n\}$. For two nonnegative functions $a(x)$ and $b(x)$ defined on the same domain, we write $a(x) \lesssim b(x)$ if there

exists an absolute constant $C > 0$ such that $a(x) \leq C\, b(x)$ for all $x$. Let $\mathcal{V}^*$ denote the set of all finite token sequences. We define two finite subsets: the initial prompt space $\mathcal{X} \subseteq \mathcal{V}^*$ and the action space $\mathcal{A} \subseteq \mathcal{V}^*$. Each action $a \in \mathcal{A}$ corresponds to a complete response represented as an autoregressively generated token sequence. A policy conditions on a given prompt or, more generally, on any sequence $h \in \mathcal{V}^*$, and is written as $\pi(\cdot \mid h)$. We use $\Pi$ to denote the class of candidate policies.

## 2 Related Work

**Parallel TTC.** Several lines of work have explored parallel TTC. Broadly, two strategies have emerged. The first is based on the self-consistency approach, where multiple answers are generated in parallel from one or more LLMs and the final output is chosen by majority vote (Wang et al., 2022; Brown et al., 2024; Chen et al., 2024). This method is simple and easy to implement but relies solely on the intrinsic ability of the LLM and often falls short of achieving the best performance. The second line of work augments parallel TTC with an external reward model, selecting the final answer according to reward scores (Song et al., 2024; Irvine et al., 2023; Puri et al., 2025). Additional parallel TTC methods introduce fine-tuning into the pipeline (Sessa et al., 2024; Chow et al., 2024). Zuo & Zhu (2025) consider allocate *across*-query allocation across different questions to further improve efficiency under fixed budgets, orthogonal to our *within*-query controller. Our work is most closely related to the second line. In contrast, we focus on improving the rate of inference through history-conditioned gating under rewards. In our experiments, we carefully control for identical verifiers and token budgets so that improvements cannot be attributed solely to stronger re-rankers, and they have been verified through multiple budget levels (Wu et al., 2024).

**Sequential TTC.** Unlike parallel TTC, sequential TTC explicitly decides *when* to spend extra steps and *where* to revise. Classical CoT techniques (Chen et al., 2023b; Zhang et al., 2024; Lee et al., 2025; Yao et al., 2023b) have been extensively studied without relying on external reward models. With reward models, several sequential TTC approaches have been explored. For example, Munkhbat et al. (2025) propose a few-shot BoN method that leverages a powerful external LLM to generate demonstrations and then selects answers via BoN. Iterative self-refinement and policy-as-verifier approaches provide alternative architectures that can be used as verifiers to trigger revisions (Madaan et al., 2023; Jiang et al., 2025). Other directions include uncertainty-aware step-wise verification (Ye et al., 2025), universal/self-consistency methods for open-ended outputs (Chen et al., 2023a; Kang et al., 2025), and "PRMs that think" (Khalifa et al., 2025), all of which further enhance the judge's

signal. By contrast, our setup belongs to this line of work but deliberately adopts a minimalist design, avoiding complex techniques such as Tree- or Graph-of-Thoughts (Yao et al., 2023a; Besta et al., 2024), while still demonstrating improved performance through sequential TTC.

**Theory of TTC.** A growing theory literature clarifies when extra samples help, when they hurt, and how to apportion TTC. For BoN, Beirami et al. (2024) correct the folklore KL identity and bound win-rate improvements, while Huang et al. (2025a) establish a coverage–error frontier that reveals BoN's reward hacking at large $N$ and propose $\chi^2$-regularized sampling with skyline-optimal, scaling-monotone guarantees. Our mechanism achieves a comparable regularization effect by *concentrating the proposal* through reward-filtered histories rather than reweighting selection. On sample complexity, Huang et al. (2025b) separate self-consistency from BoN. Foster et al. (2025) study how the base model performs in TTC in terms of coverage and the benefits of multi-turn exploration. Recently, Xu et al. (2025) analyze Learning from language/process feedback (HELIX), provide regret guarantees under latent rewards, and highlight the need for richer, process-aware signals. By contrast, our work focuses on establishing a separation result between parallel and sequential TTC, and on developing an algorithm that uses reward signals to guide sequential TTC—a direction largely absent from prior work.

## 3 Preliminaries

We assume access to a large language model (LLM) $\pi_{\text{LLM}}(\cdot \mid h)$, which receives an input sequence $h \in \mathcal{V}^*$, and then outputs a distribution over subsequent sequences. In our setting, we restrict the output sequences to lie in $\mathcal{A}$. Let $\mathcal{X}$ be the prompt set and $p_{\text{ini}} \in \Delta(\mathcal{X})$ denote a distribution over initial prompts. In each round, we first draw $x \sim p_{\text{ini}}$. Then an algorithm Alg interacts with the language model $\pi_{\text{LLM}}$ for multiple times before producing a final action $\widehat{a} \in \mathcal{A}$. We denote the resulting conditional distribution of $\widehat{a}$ by $\pi_{\text{Alg}}(\cdot \mid x)$. We assume a reward function $r : \mathcal{A} \times \mathcal{X} \to [-1, 1]$. Because the output space $\mathcal{A}$ can be extremely large, it is infeasible to explicitly evaluate either the language model $\pi_{\text{LLM}}$ or the reward function $r$ over all possible candidates. Instead, we assume that our algorithm operates under the following sequential sample-and-evaluate framework:

**Definition 3.1** (*Sequential* sample-and-evaluate framework, generalized from Huang et al. 2025a). For a given prompt $x \in \mathcal{X}$, the algorithm may sequentially issue $n$ queries $\{h_i\}_{i=1}^n$. For each $i$, it samples $a_i \sim \pi_{\text{LLM}}(\cdot \mid h_i)$,

and observes the reward value $r(a_i, x)$. The efficiency (query complexity) of the algorithm is measured by the total number of queries $n$.

*Remark* 3.2. Huang et al. (2025a) studied a more restricted setting of the sample-and-evaluate framework, where one can only sample actions from a fixed prompt $x$, i.e., from $\pi_{\text{LLM}}(\cdot \mid x)$. While conceptually simple, this framework largely ignores the internal structure and sequential nature of LLMs.

We evaluate performance with respect to a *comparator policy* $\pi^\star(\cdot|x) : \mathcal{X} \to \Delta(\mathcal{A})$. Given a prompt $x$ and comparator policy $\pi^\star$, we define the regret of an action $\widehat{a}$ as

$$\text{Regret}(\widehat{a}; x, \pi^\star) := \mathbb{E}_{a \sim \pi^\star(\cdot|x)}\big[r(a, x) - r(\widehat{a}, x)\big].$$

An action $\widehat{a}$ is $\epsilon$-optimal with respect to $\pi^*$ if $\text{Regret}(\widehat{a}; x, \pi^\star) \leq \epsilon$ for some $\epsilon \in [0, 1]$. When $\widehat{a}$ is generated by an algorithm Alg, we write $\text{Regret}(\text{Alg}; x, \pi^\star) := \mathbb{E}_{\widehat{a} \sim \pi_{\text{Alg}}(\cdot|x)}\text{Regret}(\widehat{a}; x, \pi^\star)$. Our goal is to design algorithms that, for prompts drawn from $p_{\text{ini}}$, achieve small regret with respect to $\pi^\star$ while minimizing the *sample complexity*, defined as the total number of times that the algorithm queries $\pi_{\text{LLM}}$.

## 4 Fundamental Limits of Parallel Test-Time Compute

In this section, we investigate the fundamental limits of parallel test-time computation. To this end, we impose a structural assumption on the pretraining of $\pi_{\text{LLM}}$.

**Pretraining data distribution.** We assume access to a pretraining dataset $\mathcal{D}$, consisting of trajectories collected from a reference policy $\pi_{\text{ref}}$. Formally, there exists a finite family of reference policies $\{\pi_{\text{ref}}^\tau(\cdot \mid x) : \tau \in \mathcal{T}_{\text{ref}}\}$, where $\mathcal{T}_{\text{ref}}$ is a finite index set and $p_{\text{ref}}$ denotes the prior distribution reflecting the proportion of data contributed by each reference policy. In practice, different reference policies may correspond to distinct answer-generation styles in the pretraining corpus. For instance, some may produce concise responses while others generate more elaborate or vivid explanations. These stylistic variations naturally form a finite collection of reference policies contributing to the dataset. The trajectories $h^t = (x^t, a_1^t, \ldots, a_N^t)$ in the pretraining dataset $\mathcal{D}$ are generated as follows:

- Sample an initial prompt $x^t \sim p_{\text{ini}}$ and a reference index $\tau^t \sim p_{\text{ref}}$. Initialize the history $h_0^t = x^t$.
- For $i = 1, \ldots, N$, sequentially draw an action $a_i^t \sim \pi_{\text{ref}}^{\tau^t}(\cdot \mid x^t)$ and update the history as $h_i^t = h_{i-1}^t \cup (a_i^t)$.
- Set $h^t = h_N^t$ and add it to $\mathcal{D}$.

*Remark* 4.1. In our assumption for pretraining data, all actions within the same trajectory are sampled from the same distribution $\pi_{\text{ref}}^\tau(\cdot \mid x)$. Our assumption on the pretraining distribution is aligned with those commonly made for in-context learning (Xie et al., 2021; Zhang et al., 2023). In particular, our pretraining dataset does not contain any

**Algorithm 1** Sequential Best-of-$n$ (SeqBoN)

---

**Require:** Prompt $x$, reward $r(\cdot, \cdot)$, budget $n$.
  1: **for** $i = 1, \dots, n$ **do**
  2:    Update $h_i$ based on $x, a_1, \dots, a_{i-1}$. Sample $a_i \sim \pi_{\text{LLM}}(\cdot \mid h_i)$.
  3: **end for**
**Ensure:** Return $\widehat{a} = \arg\max_{i \in [n]} r(a_i, x)$

---

reward information. This contrasts with the in-context reinforcement learning (ICRL) literature (Lin et al., 2024; Wang et al., 2024; Lee et al., 2023), which typically assumes that rewards are included in the pretraining data.

We then interpret the pretrained LLM $\pi_{\text{LLM}}$ as being trained on $\mathcal{D}$ and we call $\pi_{\text{LLM}}$ as a *mixture of reference policy model* since it consists of trajectories drawn from different reference policies. Conceptually, when the pretraining length $T$ and the number of data $N$ grow to infinity, $\pi_{\text{LLM}}$ converges to the conditional distribution induced by the data-generating process (Lee et al., 2023). We formalize this as the following assumption.

**Assumption 4.2** (Realizability of $\pi_{\text{LLM}}$). For any trajectory $h \in \mathcal{V}^*$, we have $\pi_{\text{LLM}}(\cdot \mid h) = \mathbb{P}_{\mathcal{D}}(\cdot \mid h)$, where $\mathbb{P}_{\mathcal{D}}(\cdot \mid h)$ denotes the true conditional distribution of the next action given history $h$ under the data-generating process defined by sampling $\tau \sim p_{\text{ref}}$, $x \sim p_{\text{ini}}$, and actions from $\pi_{\text{ref}}^\tau(\cdot \mid x)$.

We also make the following assumption on reward function:

**Assumption 4.3.** Given any $x \in \mathcal{X}$, $a^\star(x) := \arg\max_{a \in \mathcal{A}} r(a, x)$ is unique. Also, $r(a^\star(x), x) = 1$.

**Fundamental limits of parallel TTC.** Equipped with Assumption 4.2, we demonstrate that the vanilla Best-of-N (BoN) algorithm is suboptimal, indicating the need for a more nuanced algorithmic design. For this purpose, we first consider the sample complexity bound for the vanilla BoN algorithm, which has been studied in (Huang et al., 2025a). We will study this problem in a more general framework. To begin with, we first propose a Sequential BoN (SeqBoN) in Algorithm 1, which serves as a meta-algorithm. At each iteration $i$, Algorithm 1 will formulate an input sequence $h_i$ based on the prompt $x$ and the previous answers $a_1, \dots, a_{i-1}$, then samples a new answer $a_i$ from $\pi_{\text{LLM}}(\cdot \mid h_i)$. Apparently, Algorithm 1 takes the classical BoN algorithm as its special case when $h_i = x$ for all $i$. We first introduce $\mathcal{E}_M$-divergence and coverage.

**Definition 4.4** ($\mathcal{E}_M$-divergence and coverage, Huang et al. 2025a). Let $\pi_1, \pi_2 \in \Pi$ be two policies over the action space $\mathcal{A}$. For a rejection threshold $M \geq 1$, the $\mathcal{E}_M$-divergence between $\pi_1$ and $\pi_2$ is defined as $\mathcal{E}_M(\pi_1, \pi_2) := \sum_{a \in \mathcal{A}} \max\{0, \pi_1(a) - M\pi_2(a)\}$. We denote by $M_{\pi_1, \pi_2}^\epsilon$ the smallest $M$ such that $\mathcal{E}_M(\pi_1, \pi_2) \leq \epsilon$. We also define the *coverage* as $C(\pi_1, \pi_2) := \mathbb{E}_{a \sim \pi_1}[\pi_1(a)/\pi_2(a)]$.

Moreover, for any $0 < \epsilon < 1$, if $C(\pi_1, \pi_2) < \infty$, then the rejection threshold satisfies the upper bound $M_{\pi_1, \pi_2}^\epsilon \leq C(\pi_1, \pi_2)/\epsilon$.

First, based on Definition 4.4, we recall the sample complexity result of vanilla BoN built in Huang et al. (2025a), which is near-optimal under the parallel TTC setup.

**Proposition 4.5** (Adapted from Huang et al. 2025a). *Given $\epsilon > 0$, denote $M_{LLM}^{x, \epsilon} := M_{\pi^\star(\cdot \mid x), \pi_{LLM}(\cdot \mid x)}^\epsilon$ and $C_{LLM}^\star(x) := C(\pi^\star(\cdot \mid x), \pi_{LLM}(\cdot \mid x))$. Then vanilla BoN (Algorithm 1 with $h_i = x$) takes $n = M_{LLM}^{x, \epsilon} \cdot \log(1/\epsilon) = O(C_{LLM}^\star(x)/\epsilon)$ samples to achieve $Regret(BoN; x, \pi^\star) \leq 2\epsilon$. Meanwhile, for any parallel TTC algorithm, there exists a problem instance such that it can not return an $\epsilon$-optimal answer if $n < M_{LLM}^{x, \epsilon}$.*

Although Proposition 4.5 suggests that the optimal sample complexity for parallel TTC should be proportional to $M_{\text{LLM}}^{x, \epsilon}$, our next theorem establishes a lower bound on the sample complexity under the sequential sample-and-evaluate framework, thereby separating the parallel and sequential settings.

**Theorem 4.6** (Lower Bound of Sequential Sample-and-Evaluate Algorithms). *Suppose the comparator policy $\pi^\star(\cdot \mid x) := a^\star(x)$ is a deterministic policy. Let $\tau^\star(x) \in \mathcal{T}_{ref}$ be the reference policy index satisfying $\tau^\star(x) \in \arg\max_{\tau \in \mathcal{T}_{ref}} \log \pi_{ref}^\tau(a^\star(x) \mid x)$. Meanwhile, let $M_{\tau^\star(x)}^{x, \epsilon} := M_{\pi^\star(\cdot \mid x), \pi_{ref}^{\tau^\star(x)}(\cdot \mid x)}^\epsilon$. Then,*

- *For any sequential sample-and-evaluate algorithm $A$, there exists a reward function $r$ such that if $n < M_{\tau^\star(x)}^{x, \epsilon}$, we must have $Regret(A; x, \pi^\star) > \epsilon$.*

- *For any $\epsilon > 0$, we have $M_{\tau^\star(x)}^{x, \epsilon} \leq M_{LLM}^{x, \epsilon}$, with strict inequality whenever the reference policies $\pi_{ref}^\tau(\cdot \mid x)$ are not identical across $\tau$.*

Overall, our lower bound in Theorem 4.6 indicates that, under the sequential-type pretraining assumption of LLMs, the statistical limit achievable by any sequential-type algorithm can be strictly better than that of purely parallel methods such as vanilla BoN. This motivates the development of more efficient algorithms that explicitly leverage this revised statistical landscape.

## 5  Better Test-Time Compute with Reward-Filtered Sequences

In this section, we propose a new algorithm designed to achieve this new statistical limit. We begin with an analysis of Algorithm 1. The following theorem provides a regret bound between the comparator policy $\pi^\star$ and SeqBoN.

**Theorem 5.1.** *For any prompt $x$ and any $0 < \epsilon < 1$,*

$$Regret(SeqBoN; x, \pi^\star) \leq \epsilon$$

$$+ \mathbb{E}_{a_1,\dots,a_n} \left[ \exp \left[ - n^2 \Big/ \Big( \sum_{i=1}^{n} M_{\pi^\star(\cdot | x),\, \pi_{LLM}(\cdot | h_i)}^{\epsilon} \Big) \right] \right].$$

*Remark* 5.2. Huang et al. (2025a) showed that the vanilla BoN method satisfies Regret(BoN; $x, \pi^\star$) $\leq \epsilon + \exp(-n/M_{\text{LLM}}^{x,\epsilon})$. Our bound is tighter whenever $M_{\pi^\star(\cdot|x),\, \pi_{\text{LLM}}(\cdot|h_i)}^{\epsilon} \leq M_{\pi^\star(\cdot|x),\, \pi_{\text{LLM}}(\cdot|x)}^{\epsilon}$, which indicates that conditioning on history $h_i$ allows $\pi_{\text{LLM}}(\cdot \mid h_i)$ to provide a more accurate approximation of $\pi^\star(\cdot \mid x)$ than $\pi_{\text{LLM}}(\cdot \mid x)$.

---

**Algorithm 2** RF-SeqBoN

---

**Require:** $\pi_{\text{LLM}}, x, r(\cdot, \cdot), \gamma$
1: Initialize $\bar{h} \leftarrow \langle x \rangle$
2: **for** $i = 1, \dots, n$ **do**
3:      Set $h_i \leftarrow \bar{h}$ and sample $a_i \sim \pi_{\text{LLM}}(\cdot \mid h_i)$
4:      **If** $r(a_i, x) \geq \gamma$ **then** $\bar{h} \leftarrow \bar{h} \parallel a_i$
5: **end for**
**Ensure:** $\widehat{a} = \arg\max_{i \in [n]} r(a_i, x)$

---

Recall that, to achieve the lower bound $M_{\tau^\star(x)}^{x,\epsilon}$, it suffices to design an algorithm that constructs histories $h_i$ such that $\pi_{\text{LLM}}(\cdot \mid h_i) \to \pi_{\text{ref}}^{\tau^\star(x)}(\cdot \mid x)$. This can be accomplished if $h_i$ consists of actions drawn from the optimal reference policy $\pi_{\text{ref}}^{\tau^\star(x)}(\cdot \mid x)$. Such a strategy has been widely used in in-context learning, where a number of works establish theoretical guarantees on how many context samples are needed to achieve this convergence (Wies et al., 2023; Zhang et al., 2023; Li et al., 2023; Bai et al., 2023). However, in our setting, the reference policies $\pi_{\text{ref}}$ are inaccessible at evaluation time, since they are only available during pretraining.

We make the following observation: although we cannot directly sample from the optimal reference policy $\pi_{\text{ref}}^{\tau^\star(x)}(\cdot \mid x)$, we can instead construct $h_i$ as a *sequence of optimal actions* $a^\star(x)$, which still ensures that $\pi_{\text{LLM}}(\cdot \mid h_i) \to \pi_{\text{ref}}^{\tau^\star(x)}(\cdot \mid x)$. To see this, consider the history $h = (\underbrace{a^\star(x), \dots, a^\star(x)}_{k \text{ times}}, x)$. Given the fact that $\tau^\star(x)$ maximizes $\pi_{\text{ref}}^{\tau}(a^\star(x) \mid x)$ and $\mathbb{P}_{\mathcal{D}}(\tau^\star(x) \mid h) \to 1$ as $k \to \infty$, under our Assumption 4.2, we have

$$\pi_{\text{LLM}}(\cdot \mid h) = \sum_{\tau \in \mathcal{T}_{\text{ref}}} \pi_{\text{ref}}^{\tau}(\cdot \mid x)\, \mathbb{P}_{\mathcal{D}}(\tau \mid h)$$
$$= \sum_{\tau \in \mathcal{T}_{\text{ref}}} \pi_{\text{ref}}^{\tau}(\cdot \mid x) \frac{\pi_{\text{ref}}^{\tau}(a^\star(x) \mid x)^k\, p_{\text{ref}}(\tau)}{\sum_{\tau' \in \mathcal{T}_{\text{ref}}} \pi_{\text{ref}}^{\tau'}(a^\star(x) \mid x)^k\, p_{\text{ref}}(\tau')}$$
$$\to \pi_{\text{ref}}^{\tau^\star(x)}(\cdot \mid x).$$

Based on this observation, we investigate a simple variant of Algorithm 1, termed the *Reward-Filtered Sequential Best-of-n (RF-SeqBoN)* algorithm. At a high level, RF-SeqBoN can be viewed as a hybrid of BoN and the rewind-and-repeat strategy (Kim et al., 2024; Beirami et al., 2024). At each

iteration, the algorithm refines the answer-generation distribution by appending to the input window only those answers whose reward exceeds a threshold $\gamma$, thereby biasing the history toward near-optimal solutions. A larger $\gamma$ enforces stricter filtering at the expense of efficiency, whereas in practice a good balance is often achieved with $\gamma < 1$.

*Remark* 5.3. Our studied algorithm falls under the general selection inference (SI) framework (Creswell et al., 2022; Hu et al., 2024), where a selection module is employed to identify in-context examples. However, vanilla SI typically relies on the LLM itself, often using likelihood scores to select the most probable examples from the history. This strategy overlooks task-specific information encoded by the reward signal. In contrast, Algorithm 2 selects histories based on reward scores, thereby better capturing task-relevant information.

In our next assumption, we relax the Assumption 4.3 which requires the uniqueness of the optimal action.

**Assumption 5.4.** For any $x$, there exist a threshold $\gamma^\star(x) < 1$, a margin $\Delta(x) > 0$, and $\tau^\star(x) \in \mathcal{T}_{\text{ref}}$ such that

- The comparator policy satisfies $\text{supp}(\pi^\star(\cdot \mid x)) \subseteq \{r(a, x) \geq \gamma^\star(x)\}$.

- for any $a \in \mathcal{A}$ satisfying $r(a, x) \geq \gamma^\star(x)$, we have

$$\log \pi_{\text{ref}}^{\tau^\star(x)}(a \mid x) - \sup_{\tau \neq \tau^\star(x)} \log \pi_{\text{ref}}^{\tau}(a \mid x) \geq \Delta(x).$$

Specifically, we denote $p_{\text{LLM}}(x) := \mathbb{P}_{a \sim \pi_{\text{LLM}}(\cdot|x)}(r(a, x) \geq \gamma^\star(x))$.

Assumption 5.4 requires that any action whose reward exceeds the threshold $\gamma^\star(x)$ can be reliably attributed to the optimal reference policy $\tau^\star(x)$, in the sense that its generation probability under $\tau^\star(x)$ is at least $\Delta$ larger than under any other reference policy. This margin condition ensures that high-reward actions are statistically distinguishable from those generated by suboptimal policies, enabling the algorithm to identify $\tau^\star(x)$ with high confidence from a finite number of samples. This assumption can be satisfied under mild conditions (please refer to Appendix B.2). When it is violated, the reward feedback loop in RF-SeqBoN can in principle amplify mis-specification and lead to inference-time reward hacking; we discuss these risks and practical mitigation strategies in Appendix E.

When we have the policy coverage assumption denoted in Definition 3.1, we have the following theorem that characterizes the sample complexity of Algorithm 2.

**Theorem 5.5** (Informal)**.** *Under Assumptions 4.2 and 5.4, let $p_{LLM}(x)$ be defined as in Assumption 5.4, denote $C_{LLM}^\star(x)$ as in Proposition 4.5 and denote $\kappa(x) := \max\{p_{ref}(\tau^\star(x)), e^{-\Delta(x)}\}$. Then, under specific hyperparameter selection for $\gamma$, for sufficiently small $\epsilon > 0$,*

*Algorithm 2 finds an $\epsilon$-optimal policy to $\pi^\star$ with sample complexity*

$$n = \text{poly}\log(1/\epsilon) \cdot \kappa(x) \cdot \frac{C^\star_{LLM}(x)}{\epsilon}.$$

*Remark* 5.6. Corollary 5.5 shows that, compared with the parallel TTC with sample complexity $\frac{C^\star_{\text{LLM}}(x)}{\epsilon}$, Algorithm 4 achieves a strictly lower sample complexity since $\kappa(x) < 1$. Notably, the improvement gap will be large when the probability mass of the optimal reference policy $p_{\text{ref}}(\tau^\star(x))$ is small, yet the gap $\Delta(x)$ between the optimal and suboptimal reference policies is large.

## 6  Experiments

**Models and datasets.** Following recent work on math-reasoning evaluation under test-time scaling (Guha et al., 2025; Wang et al., 2025; Agarwal et al., 2025; Otth et al., 2025), we evaluate our approach on five benchmarks with verifiable answers, each targeting different aspects of reasoning. The first is **MATH500**, a 500-problem subset of the MATH dataset that preserves the competition-style short-answer format and topic distribution (Hendrycks et al., 2021; Lightman et al., 2023); performance is measured by exact-match accuracy on the final numeric or algebraic expression. The second is **GPQA–Diamond**, a 198-question high-agreement split of GPQA spanning biology, physics, and chemistry. This benchmark is designed to be "Google-proof," with expert-authored and validated multiple-choice questions (Rein et al., 2024); evaluation is based on standard multiple-choice accuracy. We also consider competition-style math benchmarks: **AMC'23**, a 40-problem subset of the 2023 American Mathematics Competitions assessing non-routine problem-solving in algebra, geometry, number theory, and combinatorics (Mathematical Association of America, 2023); **AIME'24**, consisting of the 30 problems from the 2024 AIME I/II, each requiring a three-digit answer in the range $[000, 999]$ (Mathematical Association of America, 2024); and **AIME'25**, analogous to AIME'24 but using the 2025 AIME I/II set (Mathematical Association of America, 2025); all of the competition-style math benchmarks are evaluated using exact-match accuracy on the final integer answer.

For foundation models, we use **Qwen3-4B-Instruct** and **Qwen3-0.6B-Thinking** as recently released open-weight, instruction-tuned models which exhibits competitive capabilities at small scales, including instruction following, multilingual usage, and reasoning, and are broadly available under permissive licenses (Yang et al., 2025b;a). In addition, we use **GPT5-Nano**, a recently released closed-weight reasoning model optimized for low test-time compute and latency, which is accessed exclusively through an API-based deployment (Singh et al., 2025). Together, these open- and closed-weight models enable evaluation under diverse capac-

ities and deployment constraints, providing a comprehensive test of the robustness and generality of our algorithms.

Reward models (RMs) map candidate solutions to scalar signals. We use a Process Reward Model (PRM): **Llama3.1-8B-PRM-Deepseek-Data**, where we only use the aggregation reward for each answer we generate once, a scalar reward $r \in [0, 1]$ (RLHFlow, 2024).

**Baselines and implementation.** We compare **RF-SeqBoN** implemented following Algorithm 2 against several variants of Algorithm 1 to suggest the validity of the reward-filtering scheme. In detail, we compare Algorithm 2 with **Vanilla BoN (BoN)**, which refers to Algorithm 1 with $h_i = \langle x \rangle$; **Pure Sequential BoN (PureSeq)**: A single trajectory with a fixed revision schedule and no cross-trajectory selection, which refers to Algorithm 1 with $h_i = \langle x, a_1, \ldots, a_{i-1} \rangle$. In practice, due to the maximum token limit of LLMs, for PureSeq and RF-SeqBoN we restrict the number of appended answers in $h_i$ to a fixed budget, maintaining the context as a sliding window of most recent generations in a first-in-first-out (FIFO) manner. Details are deferred to Appendix D.1.

**Results.** We characterize accuracy–compute trade-offs across datasets and models. Accuracy is measured via exact-match (for MATH500, AMC'23, AIME'24 and AIME'25) and multiple-choice accuracy (for GPQA–Diamond). Figure 1 summarizes accuracy as a function of total test-time budget $N$ for BoN, PureSeq, and our method RF-SeqBoN across the three benchmarks. We have the following observations:

- RF-SeqBoN, which conditions the input on a history filtered by reward scores, consistently improves over vanilla BoN. This validates that maintaining a reward-biased history provides more informative context and leads to higher final accuracy. PureSeq also leverages history, but it keeps all past generations without filtering. As a result, it accumulates both useful and noisy answers, which makes it less stable than RF-SeqBoN on GPQA-Diamond and AIME'25 with larger budgets.

- As $N$ increases, we observe two regimes. On benchmarks like MATH500 and AIME'24, the advantage of RF-SeqBoN persists or even grows with larger budgets. In contrast, on GPQA-Diamond, AMC'23 and AIME'25, the baselines gradually catch up at high budgets, suggesting that all methods are approaching the intrinsic limit of the backbone model rather than continuing to benefit from more trajectories.

- With smaller models (Qwen3-0.6B-Thinking and GPT5-Nano), absolute performance is lower, but RF-SeqBoN still consistently outperforms both BoN and PureSeq, suggesting that the reward-filtered history mechanism is ro-

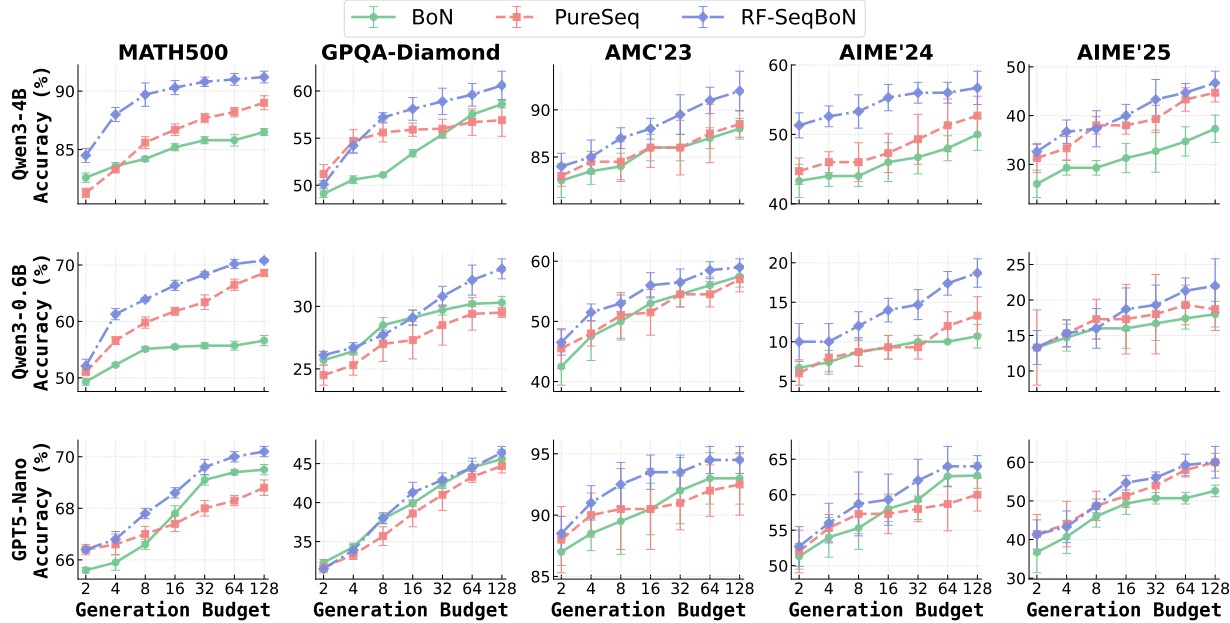

*Figure 1.* Main results with Qwen3-4B-Instruct (**top**), Qwen3-0.6B-Thinking (**medium**) and GPT5-Nano (**bottom**) foundation models of values with generation budget $N$. Each column corresponds to one benchmark dataset. The points and error bars show the mean and standard deviation across five repeated experiments, respectively.

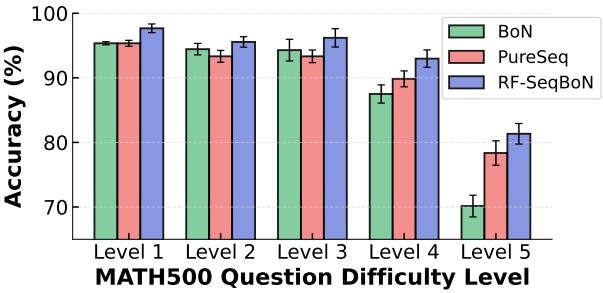

*Figure 2.* Performance breakdown on MATH500 at generation budget $N = 128$ across five question difficulty levels. Bars denote the mean accuracy, while error bars indicate the standard deviation over five repeated experiments.

bust across model scales.

**Case study.** We conduct a breakdown analysis of MATH500 by difficulty levels. We follow the level information brought by Hendrycks et al. (2021) ranging from 1 (easiest) to 5 (most challenging). The results of generation budget $N = 128$ are presented in Figure 2, using the Qwen3-4B-Instruct foundation model, consistent with the main results reported in Figure 1. We observe a clear trend of decreasing accuracy as the problem difficulty increases.

In particular, PureSeq performs slightly worse than BoN on easier subsets (levels 1–3), but achieves substantial gains on more difficult subsets (levels 4–5). Across all difficulty levels, RF-SeqBoN consistently outperforms both baselines, demonstrating its robustness in handling problems of varying complexity.

## 6.1 Ablation Studies

**Choice of hyperparameter $\gamma$.** To quantify the effect of the reward–filter threshold $\gamma$, we evaluate RF-SeqBoN with Qwen3-4B-Instruct on MATH500 while varying $\gamma$. Figure 3(a) reports accuracies across generation budgets. We observe that the choice of $\gamma$ indeed affects the performance of RF-SeqBoN. If $\gamma$ is too low, noisy trajectories persist and hinder self-refinement; if $\gamma$ is too high (e.g., $\gamma \approx 1$), most candidates are rejected and RF-SeqBoN degenerates toward BoN, forfeiting the benefits of using history for refinement. Meanwhile, our method is not particularly sensitive to $\gamma$ within a reasonable range: the performance for $\gamma = 0.93, 0.95, 0.97$ remains nearly identical and optimal, while only $\gamma = 0.9$ and $\gamma = 0.99$ exhibit noticeable drops. Additional statistics on how many filtered answers remain in the LLM context for each value of $\gamma$ at $N = 128$ on MATH-500 are reported in Appendix D.5.

**Choice of reward models.** We also adapt an outcome reward model (ORM), **AceMath-7B-RM**, to assess the robustness of reward signals (Liu et al., 2024). Compared with the PRM, this provides a weaker reward signal since intermediate reasoning steps are not evaluated. Figure 3(b) reports accuracies across generation budgets under this ORM configuration. We find that RF-SeqBoN still remains the best among the baselines, validating the effectiveness of the reward-filtering strategy under different reward models. However, unlike with PRM, our adapted ORM exhibits a minor performance drop when the budget is large. We suspect this phenomenon may be attributed to *reward over-*

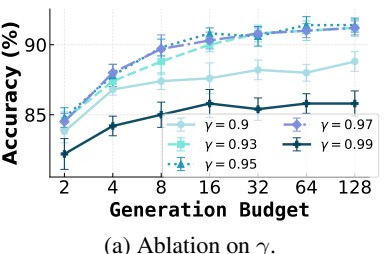

(a) Ablation on $\gamma$.

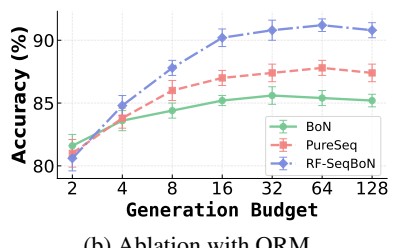

(b) Ablation with ORM.

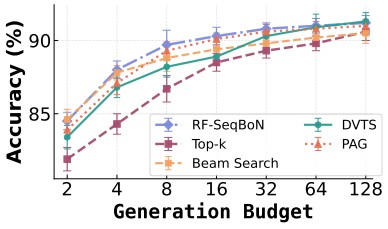

(c) Comparison with Other Baseline Algorithms.

*Figure 3.* Ablation study results on MATH500 using Qwen3-4B-Instruct as the foundation model. Points denote the mean performance, with error bars representing the standard deviation over five repeated experiments.

*optimization* (Gao et al., 2023; Frick et al., 2024; Huang et al., 2025a), and how to incorporate the reward-filtering idea to mitigate reward over-optimization remains an interesting future direction.

**Comparison with other baseline algorithms.** In addition to the BoN and PureSeq baselines, we further compare RF-SeqBoN with several stronger sequential and verifier-guided test-time compute baselines, including **Top-$k$**, **Beam Search**, **DVTS**, and **PAG**. The Top-$k$ refinement strategy does not use an explicit reward threshold. Instead, for each question, it maintains all previously generated answers and their reward scores, and constructs the history $h_i$ by appending the top-$k$ responses ranked by reward, subject to the same sliding-window truncation used for PureSeq and RF-SeqBoN. To ensure a fair comparison, we set $k = 3$, matching the `history_budget=3` used for all sequential TTC algorithms (see Table 1 in Appendix D.1), so that Top-$k$ and RF-SeqBoN differ only in whether history is defined by a relative ranking or by crossing a fixed reward threshold $\gamma$. We also include three stronger baselines: Beam Search (Snell et al., 2024), a verifier-guided tree-search method adapted to the same scalar reward interface as RF-SeqBoN; DVTS (Beeching et al., 2024), which improves search diversity by partitioning the search budget into multiple independent subtrees; and PAG (Jiang et al., 2025), a sequential self-correction method in which the model alternates between policy generation and generative verification. For Beam Search and DVTS, we set the beam-width parameter to $M = 4$ whenever $N \geq 4$, and use $M = N$ for the smallest budget $N = 2$ to respect the matched generation budget. For PAG, we use the same capped context window as the other sequential methods, with `history_budget=3`, so that at most three previous attempts or revisions are retained in the prompt. All comparisons are conducted under a matched token-budget setting on the MATH500 dataset, using Qwen3-4B-Instruct as the backbone model and the same Llama3.1-8B-PRM-Deepseek-Data reward model configuration as above. To ensure an apple-to-apple comparison, the PRM is used through the same final aggregated scalar reward for candidate ranking and selection, rather than provid-

ing Beam Search or DVTS with additional intermediate-step supervision.

Figure 3(c) reports accuracy as a function of the generation budget $N$. Across the full range of budgets, RF-SeqBoN consistently outperforms Top-$k$, especially in the low and middle budget regimes. This supports the intuition that relative ranking alone is insufficient for constructing a reliable sequential history. When genuinely high-quality solutions are rare, the top-$k$ pool may still contain several responses that are only locally better than the other sampled candidates but remain incorrect or logically flawed. Once these responses are repeatedly recycled into the context, the method can reinforce an erroneous reasoning trajectory and reduce the benefit of further sequential refinement. In contrast, RF-SeqBoN uses an absolute reward threshold to filter the history, discarding all responses whose reward falls below $\gamma$. This mechanism prevents low-quality candidates from entering the prompt even if they are ranked near the top within a weak candidate pool, thereby making each history slot more selective and informative.

RF-SeqBoN also remains highly competitive with Beam Search, DVTS, and PAG, despite using a substantially simpler single sequence reward-filtering mechanism rather than explicit tree search or self-revision. Beam Search and DVTS allocate computation to branching and verifier-guided expansion over multiple search paths, while PAG introduces an additional verify-then-revise procedure. These methods can benefit from richer search structures, especially when the budget is large enough to support diverse exploration. Nevertheless, RF-SeqBoN matches or exceeds these baselines at larger budgets, suggesting that carefully filtering and reusing high-reward trajectories can recover much of the benefit of more complex search strategies. The comparison also highlights an important practical advantage of RF-SeqBoN: it improves sequential test-time compute by modifying only the history construction rule, without requiring specialized tree search, separate branching heuristics, or additional verifier-generation roles. These results demonstrate that RF-SeqBoN offers a favorable tradeoff between accuracy, simplicity, and implementation cost among both

standard sequential baselines and stronger verifier-guided test-time compute algorithms.

## 7 Conclusion

We presented a principled framework for TTC under a mixture-of-policies model and derived fundamental limits on the budget required for near-optimal performance. Our analysis shows that parallel TTC strategies such as BoN are inherently suboptimal. To address this, we proposed *Reward-Filtered Sequential Best-of-n (RF-SeqBoN)*, which selectively incorporates high-reward generations and provably refines the distribution toward the optimal policy. Experiments across diverse benchmarks confirm that RF-SeqBoN offers both stronger theoretical guarantees and substantial empirical gains in budget efficiency.

## Acknowledgments

We thank the anonymous reviewers and the area chair for their helpful comments. We are grateful to Prof. David Crandall for his guidance and feedback on this work, and to Qingyue Zhao for active discussions during the early stages of this project. QD and QG are supported in part by the National Science Foundation under grants DMS-2323113, IIS-2403400, and CPS-2312094, as well as by a Sloan Research Fellowship. The views and conclusions expressed in this paper are those of the authors and should not be interpreted as representing the views of any funding agencies. DZ thanks Lambda, Inc. for supporting this research with cloud computing credits.

## Impact Statement

This work studies test-time compute strategies for large language models, proposing a reward-filtered sequential inference method with theoretical guarantees and empirical validation on benchmark reasoning datasets. The research is methodological and analytical in nature: it does not involve human subjects, personal data, or sensitive information, and all experiments are conducted using publicly available benchmarks and simulator-style evaluation protocols. While our results demonstrate improved efficiency and robustness in inference-time decision making, any deployment of such methods in real-world or safety-critical applications (e.g., education, healthcare, or finance) should be accompanied by careful validation of reward models, monitoring for reward mis-specification or over-optimization, and appropriate domain-specific safeguards.

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

## Contents of the Appendix

# A    Proof in Section 4

In this section, we provide proofs in Section 4.

## A.1    Proof of Proposition 4.5

*Proof of Proposition 4.5.*  Using Lemma F.1 in Huang et al. (2025a), for any $M > 0$, the regret of vanilla BoN can be upper bounded by

$$\text{Regret}(\text{BoN}, x, \pi^*) \le \mathcal{E}_M(\pi^*, \pi_{\text{LLM}}) + \exp\Big(-\frac{n}{M} \cdot \big(1 - \mathcal{E}_M(\pi^*, \pi_{\text{LLM}})\big)\Big).$$

When taking $M = M_{\text{LLM}}^{x,\epsilon}$, we have $\mathcal{E}_M(\pi^*, \pi_{\text{LLM}}) \le \epsilon$. Therefore, taking $n = M_{\text{LLM}}^{x,\epsilon} \log(1/\epsilon)$ will lead to Regret$(\text{BoN}, x, \pi^*) \le 2\epsilon$. On the other hand, for any parallel TTC algorithm $A$, consider the following problem instance, which is the same as that in the proof of Theorem 4.6. Let the reward be defined as:

$$r(a) := \begin{cases} 1, & \pi^\star(a) \ge \pi_A(a), \\ -1, & \pi^\star(a) < \pi_A(a). \end{cases}$$

Therefore, the regret of algorithm $A$ can be represented as the TV distance between $\pi^*$ and $\pi_A$, i.e.,

$$\begin{aligned}
\text{Regret}(A; x, \pi^\star) &= \mathbb{E}_{a \sim \pi^\star(\cdot)} r(a) - \mathbb{E}_{a \sim \pi_A(\cdot)} r(a) \\
&= \sum_{a \in \mathcal{A}} r(a)(\pi^\star(a) - \pi_A(a)) \\
&= \sum_{a \in \mathcal{A}} |\pi^\star(a) - \pi_A(a)|.
\end{aligned} \tag{A.1}$$

Using Theorem D.6 in Huang et al. (2025a), when the number of samples $n \le M_{\text{LLM}}^{x,\epsilon}$, the TV distance between $\pi^*$ and $\pi_A$ can be lower bounded by $\epsilon$.

$$\text{TV}\big(\pi^*(\cdot|x), \pi(\cdot|x)\big) \ge \epsilon. \tag{A.2}$$

Combining (A.1) and (A.2), we have

$$\text{Regret}(A; x, \pi^\star) \ge \epsilon.$$

This completes the proof of Proposition 4.5.                                                                        □

## A.2    Proof of Theorem 4.6

We omit $x$ in the following proof. Denote $\tau$ such that $\tau$ is the index that takes the maximum action with highest probability, i.e.,

$$\tau^\star(x) \in \arg \max_{\tau \in \mathcal{T}_{\text{ref}}} \log \pi_{\text{ref}}^\tau(a^\star(x) \mid x). \tag{A.3}$$

Then fix any $M$, denote

$$\mathcal{A}_M := \{a : \pi^\star(a) \ge M \cdot \pi^\tau(a)\}. \tag{A.4}$$

*Proof of Theorem 4.6.*  For the first result, we set the reward $r$ as

$$r(a) := \begin{cases} 1, & \pi^\star(a) \ge \pi_A(a), \\ -1, & \pi^\star(a) < \pi_A(a), \end{cases}$$

Then we have

$$\text{Regret}(A; x, \pi^\star) = \mathbb{E}_{a \sim \pi^\star(\cdot)} r(a) - \mathbb{E}_{a \sim \pi_A(\cdot)} r(a)$$

---

**Algorithm 3** Adaptive Rejection Sampling ($\texttt{RejectionSampling}_n((w_i, p_i, M_i)_{i=1}^n, x)$)

---

**Require:** Prompt $x$, sampling budgets $n$, adaptive sampling policies $(p_i)_{i=1}^n$, importance weights $(w_i)_{i=1}^n$, truncation level $(M_i)_{i=1}^n$.

1: Sample $(a_1, \ldots, a_n, a_{n+1})$ satisfying $a_i \sim p_i$, where $p_i(\cdot)$ can depend on $a_1, \ldots, a_{i-1}$.
2: **for** $i = 1 \ldots n$ **do**
3:   Sample a Bernoulli random variable $\xi_i$ such that $\mathbb{P}(\xi_i = 1 \mid a_i) = q_i, q_i := \min\left\{\frac{w_i(a_i)}{M_i}, 1\right\}$, where $w_i, M_i$ can depend on $a_1, \ldots, a_{i-1}$.
4:   **if** $\xi_i = 1$ **then**
5:     Return $a_i$
6:   **end if**
7: **end for**
8: Return $a_{n+1}$

---

$$
\begin{aligned}
&= \sum_{a \in \mathcal{A}} r(a)(\pi^\star(a) - \pi_A(a)) \\
&= \sum_{a \in \mathcal{A}} |\pi^\star(a) - \pi_A(a)|.
\end{aligned}
$$

In this example, the regret of algorithm $A$ with respect to $\pi^\star$ is equivalent to the total variation distance between $\pi^\star$ and $\pi_A$. Next, if $n < M_{\tau^\star(x)}^{x,\epsilon}$, we have

$$
\begin{aligned}
\sum_{a \in \mathcal{A}} |\pi^\star(a) - \pi_A(a)| &\geq \sum_{a \in \mathcal{A}_M} |\pi^\star(a) - \pi_A(a)| \\
&\geq \sum_{a \in \mathcal{A}_M} \pi^\star(a) - \mathbb{E}_{a_i \sim p_i} \mathbb{P}(a \in \{a_i\}) \\
&\geq \sum_{a \in \mathcal{A}_M} \pi^\star(a) - n \cdot \pi_{\text{ref}}^\tau(a \mid x) \\
&= \mathcal{E}_n(\pi^\star, \pi_{\text{ref}}^\tau(\cdot \mid x)),
\end{aligned}
\tag{A.5}
$$

where the second inequality we use the fact that $\pi_A$ can only output $a$ when $a \in \{a_i\}$, $p_i(\cdot) = \pi_{\text{LLM}}(\cdot | h_i)$. The third one holds due to the fact that for any $a \in \mathcal{A}_M \subseteq \text{supp}(\pi^\star)$, we have $\pi_{\text{ref}}^\tau(a)$ achieves the maximum probability over $\tau \in \mathcal{T}_{\text{ref}}$, i.e., $\pi_{\text{ref}}^\tau(a) \geq p_i(a)$. Therefore, recall that $n < M_{\tau^\star(x)}^{x,\epsilon}$, and $M_{\tau^\star(x)}^{x,\epsilon}$ is the smallest $M$ such that $\mathcal{E}_M(\pi^\star, \pi_{\text{ref}}^\tau(\cdot \mid x)) \leq \epsilon$. We have $\text{Regret}(A; x, \pi^\star) \geq \epsilon$.

For the second result, we show that $M_{\tau^\star(x)}^{x,\epsilon} \leq M_{\text{LLM}}^{x,\epsilon}$. By definition of the $\mathcal{E}_M$-divergence (Definition 4.4), we have

$$
\begin{aligned}
\mathcal{E}_{M_{\text{LLM}}^{x,\epsilon}}(\pi^\star(\cdot \mid x), \pi_{\text{ref}}^{\tau^\star(x)}(\cdot \mid x)) &:= \sum_{a \in \mathcal{A}} \max\{0, \pi^\star(\cdot \mid x) - M_{\text{LLM}}^{x,\epsilon} \pi_{\text{ref}}^{\tau^\star(x)}(\cdot \mid x)\} \\
&\leq \sum_{a \in \mathcal{A}} \max\{0, \pi^\star(\cdot \mid x) - M_{\text{LLM}}^{x,\epsilon} \pi_{\text{LLM}}(\cdot \mid x)\} \\
&\leq \epsilon,
\end{aligned}
$$

where the first inequality holds due to (A.3). The second inequality holds due to the definition of $M_{\text{LLM}}^{x,\epsilon}$. Using again $M_{\tau^\star(x)}^{x,\epsilon}$ is the smallest $M$ such that $\mathcal{E}_M(\pi^\star(\cdot \mid x), \pi_{\text{ref}}^\tau(\cdot \mid x)) \leq \epsilon$, we draw the final conclusion. $\square$

## B  Proof in Section 5

We first prove Theorem 5.1. In order to prove that, we first propose a generalized version of rejection

### B.1  Proof of Theorem 5.1

First, we consider the generalized version of rejection sampling in Algorithm 3 as an auxiliary policy. Specifically, we set the importance weights $w_i(a) := \pi^\star(a)/p_i(a)$. The result is formulated in the following theorem.

**Theorem B.1.** *For any comparator policy $\pi^\star$, let $\pi_R$ denote the distribution induced by the adaptive rejection sampling algorithm* $\texttt{RejectionSampling}_n((w_i, p_i, M_i)_{i=1}^n, x)$ *(Algorithm 3) with importance weights $w_i(a) := \pi^\star(a)/p_i(a)$. Then the total variation distance between $\pi^\star$ and $\pi_R$ can be upper bounded by:*

$$\mathrm{TV}(\pi^\star, \pi_R) \;\leq\; \sum_{i=1}^n \alpha_i \mathbb{E}_{a_1,\ldots,a_n}[\mathcal{E}_i] \;+\; 2\alpha_{n+1},$$

*where $\mathcal{E}_i := \mathcal{E}_{M_i}(\pi^\star, p_i)$, and $(\alpha_i)_{i=1}^{n+1}$ is a sequence of positive weights satisfying $\sum_{i=1}^{n+1} \alpha_i = 1$. Furthermore, we have*

$$\alpha_{n+1} \leq \mathbb{E}_{a_1,\ldots,a_N}\left[\exp\left(-\sum_{i=1}^n \frac{1}{M_i}\right)\right].$$

*If we set $M_i = M^\epsilon_{\pi^\star(\cdot,x), \pi_{LLM}(\cdot|h_i)}$, we have $\mathcal{E}_i \leq \epsilon$ due to Definition 4.4. Then the TV distance can be bounded by*

$$\mathrm{TV}(\pi^\star, \pi_R) \leq \epsilon + \mathbb{E}_{a_1,\ldots,a_n}\left[\exp\left(-\sum_{i=1}^n \frac{1}{M^\epsilon_{\pi^\star(\cdot|x), \pi_{LLM}(\cdot|h_i)}}\right)\right]. \tag{B.1}$$

*Proof.* For simplicity, we fix $x$ and ignore all the $x$ dependence. We denote $p_i(\cdot) := \pi_{\mathrm{LLM}}(\cdot \mid h_i)$. Following the proof of Lemma D.4 in Huang et al. (2025a), we consider the truncated pseudo-distribution

$$\pi_i(a) := \min\{\pi^\star(a), M_i p_i(a)\},$$

and $A_i := \sum_a \pi_i(a) \leq 1$ is the total mass of $\pi_i$. Moreover, we have

$$\begin{aligned}
A_i &= \sum_a \min\{\pi^\star(a), M_i p_i(a)\} \\
&= \sum_{a \notin \mathcal{A}_i} \pi^\star(a) + \sum_{a \in \mathcal{A}_i} M_i p_i(a) \\
&= 1 - \sum_{a \in \mathcal{A}_i} \left[\pi^\star(a) - M_i p_i(a)\right].
\end{aligned}$$

where $\mathcal{A}_i = \{a \in \mathcal{A} : \pi^*(a) > M_i p_i(a)\}$. Recalling the definition of $\mathcal{E}_{M_i}(\pi^\star, p_i)$ in Definition 4.4, we have

$$\mathcal{E}_i := \mathcal{E}_{M_i}(\pi^\star, p_i) = 1 - A_i.$$

Following the proof of Lemma D.4 in Huang et al. (2025a), we have

$$\mathbb{P}_{a' \sim p_i}(a' = a \mid \xi_i = 1) = \frac{\pi_i(a)}{A_i}.$$

From now on, we begin to prove the original statement. For $a_i \sim p_i$, let $q_i := \min\left\{\frac{\pi^\star(a_i)}{M_i p_i(a_i)}, 1\right\}$ be the probability of acceptance at step $i$. We define

$$\alpha_i := \mathbb{E}_{a_1,\ldots,a_i}\left[(1-q_1)\cdots(1-q_{i-1})q_i\right], \quad \alpha_{n+1} := \mathbb{E}_{a_1,\ldots,a_n}\left[(1-q_1)\cdots(1-q_n)\right],$$

so that $\sum_{i=1}^{n+1} \alpha_i = 1$. Let $i^\star$ denote the index at which Algorithm 3 stops, and let $\widehat{a}$ denote the output. Hence,

$$\begin{aligned}
\pi_R(a) &= \mathbb{P}(\widehat{a} = a) \\
&= \underbrace{\sum_{i=1}^N \mathbb{P}(\widehat{a} = a \mid i^\star = i)\mathbb{P}(i^\star = i)}_{T_1} + \underbrace{\mathbb{P}(\widehat{a} = a \mid i^\star = n+1)\mathbb{P}(i^\star = n+1)}_{T_2}. \tag{B.2}
\end{aligned}$$

For the first term $T_1$, we compute

$$
\begin{aligned}
T_1 &= \sum_{i=1}^{n} \mathbb{E}_{a_1,\ldots,a_{i-1}} \mathbb{P}_{a' \sim p_i}(a' = a \mid \xi_i = 1) \, \mathbb{E}_{a_1,\ldots,a_n} \mathbb{P}(i^\star = i \mid a_1,\ldots,a_n) \\
&= \sum_{i=1}^{n} \mathbb{E}_{a_1,\ldots,a_{i-1}} \frac{\pi_i(a)}{A_i} \cdot \underbrace{\mathbb{E}_{a_1,\ldots,a_i}(1-q_1)\cdots(1-q_{i-1})q_i}_{\alpha_i} \\
&= \sum_{i=1}^{n} \alpha_i \mathbb{E}_{a_1,\ldots,a_n} \frac{\pi_i(a)}{A_i} \\
&= \mathbb{E}_{a_1,\ldots,a_n}\left[\sum_{i=1}^{n} \alpha_i \frac{\pi_i(a)}{A_i}\right].
\end{aligned}
\tag{B.3}
$$

For the second term $T_2$, we have

$$
\begin{aligned}
T_2 &\leq \mathbb{P}(i^\star = n+1) \\
&= \mathbb{E}_{a_1,\ldots,a_n} \mathbb{P}(i^\star = n+1 \mid a_1,\ldots,a_n) \\
&= \underbrace{\mathbb{E}_{a_1,\ldots,a_n}(1-q_1)\cdots(1-q_n)}_{\alpha_{n+1}}.
\end{aligned}
\tag{B.4}
$$

Substituting (B.3) and (B.4) into (B.2), we have

$$
\begin{aligned}
|\pi^\star(a) - \pi_R(a)| &\leq |\pi^\star(a) - T_1| + T_2 \\
&\leq \left|\sum_{i=1}^{n+1} \alpha_i \pi^\star(a) - \mathbb{E}_{a_1,\ldots,a_n}\left[\sum_{i=1}^{n} \alpha_i \frac{\pi_i(a)}{A_i}\right]\right| + \alpha_{n+1} \\
&\leq \left|\sum_{i=1}^{n} \alpha_i \pi^\star(y) - \mathbb{E}_{a_1,\ldots,a_n}\left[\sum_{i=1}^{n} \alpha_i \frac{\pi_i(a)}{A_i}\right]\right| + 2\alpha_{n+1},
\end{aligned}
$$

where the second inequality uses $\sum_{i=1}^{n+1} \alpha_i = 1$. Now consider the total variation distance:

$$
\begin{aligned}
\mathrm{TV}(\pi^\star, \pi_R) &:= \sum_{a} |\pi^\star(a) - \pi_R(a)| \\
&\leq \sum_{a} \left|\sum_{i=1}^{n} \alpha_i \pi^\star(a) - \mathbb{E}_{a_1,\ldots,a_n}\left[\sum_{i=1}^{n} \alpha_i \frac{\pi_i(y)}{A_i}\right]\right| + 2\alpha_{n+1} \\
&\leq \sum_{a} \left|\mathbb{E}_{a_1,\ldots,a_n}\left[\sum_{i=1}^{n} \alpha_i \left(\pi^\star(a) - \tfrac{\pi_i(a)}{A_i}\right)\right]\right| + 2\alpha_{n+1} \\
&\leq \sum_{i=1}^{n} \alpha_i \mathbb{E}_{a_1,\ldots,a_n}\left[\sum_{a} \left|\pi^\star(a) - \tfrac{\pi_i(a)}{A_i}\right|\right] + 2\alpha_{n+1},
\end{aligned}
\tag{B.5}
$$

where the last inequality holds due to the triangle inequality. Finally, for each $i$, we bound

$$
\begin{aligned}
\sum_{a} \left|\pi^\star(a) - \tfrac{\pi_i(a)}{A_i}\right| &\leq \sum_{a} |\pi^\star(a) - \pi_i(a)| + \sum_{a} \left|\pi_i(a) - \tfrac{\pi_i(a)}{A_i}\right| \\
&= \sum_{a \in \mathcal{A}_i} (\pi^\star(a) - M_i p_i(a)) + \sum_{a} \left(\tfrac{\pi_i(a)}{A_i} - \pi_i(a)\right) \\
&= \mathcal{E}_i + (1 - A_i) \\
&= 2\mathcal{E}_i,
\end{aligned}
\tag{B.6}
$$

where $\mathcal{A}_i = \{a \in \mathcal{A} : \pi^*(a) > M_i p_i(a)\}$. The first equation holds due to $\pi_i(a) := \min\{\pi^\star(a), M_i p_i(a)\}$ and $A_i \leq 1$. The second equation holds due to $\sum_a \pi_i(a) = A_i$. The last equation holds due to $\mathcal{E}_i = 1 - A_i$. Substituting (B.6) into (B.5)

yields

$$\mathrm{TV}(\pi^\star, \pi_R) \leq \sum_{i=1}^{n} \alpha_i \mathbb{E}_{a_1,\ldots,a_n}[\mathcal{E}_i] + 2\alpha_{n+1}. \tag{B.7}$$

Moreover, we have

$$\alpha_{n+1} = \mathbb{E}_{a_1,\ldots,a_n}\left[\prod_{i=1}^{n}\left(1 - \frac{A_i}{M_i}\right)\right] = \mathbb{E}_{a_1,\ldots,a_n}\left[\prod_{i=1}^{n}\left(1 - \frac{1-\mathcal{E}_i}{M_i}\right)\right]. \tag{B.8}$$

Finally, setting $M_i = M^\epsilon_{\pi^\star(\cdot|x),\pi_{\mathrm{LLM}}(\cdot|h_i)}$ we have $\mathcal{E}_i \leq \epsilon$, and

$$\begin{aligned}
\alpha_{n+1} &= \mathbb{E}_{a_1,\ldots,a_n}\left[\prod_{i=1}^{n}\left(1 - \frac{1-\mathcal{E}_i}{M_i}\right)\right] \\
&\leq \mathbb{E}_{a_1,\ldots,a_n}\left[\exp\left(-\sum_{i=1}^{n}\frac{1}{M_i}\right)\right] \\
&\leq \mathbb{E}_{a_1,\ldots,a_n}\left[\exp\left(-\frac{n^2}{\sum_{i=1}^{n} M^\epsilon_{\pi^\star(\cdot|x),\pi_{\mathrm{LLM}}(\cdot|h_i)}}\right)\right]
\end{aligned} \tag{B.9}$$

where we use $1 - x \leq \exp(-x), \forall x \in \mathbb{R}$ and Definition 4.4. Combining (B.8) and (B.9), we complete the proof of Theorem B.1. $\qquad\square$

Using this result, now we can prove Theorem 5.1.

*Proof of Theorem 5.1.* let $\pi_R$ be the auxilliary distribution defined in Theorem B.1. To begin with, we have

$$\begin{aligned}
&\mathbb{E}_{a\sim\pi^\star}[r(a,x)] - \mathbb{E}_{a\sim\pi_{\mathrm{abon}}}[r(a,x)] \\
&= \underbrace{\mathbb{E}_{a\sim\pi^\star}[r(a,x)] - \mathbb{E}_{a\sim\pi_R}[r(a,x)]}_{I_1} + \underbrace{\mathbb{E}_{a\sim\pi_R}[r(a,x)] - \mathbb{E}_{a\sim\pi_{\mathrm{abon}}}[r(a,x)]}_{I_2}.
\end{aligned}$$

For $I_1$, we have

$$I_1 \leq \mathrm{TV}(\pi^\star, \pi_R).$$

For $I_2$, we have

$$I_2 = \mathbb{E}_{a_1,\ldots,a_n}\left[\mathbb{E}_{a\sim\pi_R|a_1,\ldots,a_n}[r(a,x)] - \mathbb{E}_{a\sim\pi_{\mathrm{abon}}|a_1,\ldots,a_n}[r(a,x)]\right].$$

Note that for any fixed $a_1,\ldots,a_n$, Algorithm 1 will always return $a_i$ which achieves the maximum reward. Thus, $I_2$ is non-positive. Finally, we use Theorem B.1 to bound $\mathrm{TV}(\pi^\star, \pi_R)$ and thus we complete the proof of Theorem 5.1. $\qquad\square$

## B.2 Proof of Proposition B.2

**Proposition B.2.** *Suppose the following conditions hold for any $x$: there exists some embedding $\phi(a, x)$ such that*

- *The reward function $r(\cdot, x)$ is smooth at $a^\star(x)$ w.r.t. $\phi(a, x)$.*

- *For any $\tau \in \mathcal{T}_{\mathrm{ref}}$, the reference policy $\log \pi^\tau_{\mathrm{ref}}(\cdot \mid x)$ is $L$-smooth w.r.t. $\phi(a, x)$.*

- *There exists $\Delta(x) > 0$ such that $\log \pi^{\tau^\star(x)}_{\mathrm{ref}}(a^\star(x) \mid x) - \sup_{\tau \neq \tau^\star(x)} \log \pi^\tau_{\mathrm{ref}}(a^\star(x) \mid x) > \Delta(x)$.*

*Then there exists a threshold $\gamma^\star(x) < 1$ such that Assumption 5.4 holds.*

Proposition B.2 provides a sufficient condition for Assumption 5.4. The smoothness of $r$ and $\pi_{\text{ref}}$ ensures that in a neighborhood of $a^\star(x)$, both the reward value and the probability gap between $\tau^\star(x)$ and other reference policies vary continuously. Since the margin at $a^\star(x)$ is strictly positive ($> \Delta(x)$), this gap persists in a small neighborhood around $a^\star(x)$. Consequently, there exists a reward threshold $\gamma^\star(x) < 1$ such that any action with $r(a, x) \geq \gamma^\star(x)$ lies within this neighborhood and inherits the same probability margin. Intuitively, if the optimal action is well separated from all others in terms of generation probability, and both the reward and policy distributions change smoothly, this separation extends to all sufficiently high-reward actions.

*Proof of Proposition B.2.* Since $r$ is smooth on the point $a^\star(x)$, then there must exists $\gamma^*(x)$ such that

$$r(a^\star(x), x) - r(a, x) < 1 - \gamma^*(x) \Rightarrow \|\phi(a, x) - \phi(a^\star(x), x)\| \leq \Delta/(4L). \tag{B.10}$$

Then, due to smoothness assumption on $\pi_{\text{ref}}$, we have

$$\log \pi_{\text{ref}}^{\tau^\star(x)}(a \mid x) - \sup_{\tau \neq \tau^\star(x)} \log \pi_{\text{ref}}^\tau(a \mid x)$$

$$\geq \log \pi_{\text{ref}}^{\tau^\star(x)}(a^\star(x) \mid x) - L\|\phi(a, x) - \phi(a^\star(x), x)\|$$

$$- \sup_{\tau \neq \tau^\star(x)} \left( \log \pi_{\text{ref}}^\tau(a^\star(x) \mid x) \right) - L\|\phi(a, x) - \phi(a^\star(x), x)\|$$

$$> \Delta - 2L\|\phi(a, x) - \phi(a^\star(x), x)\|$$

$$\geq \Delta/2.$$

$\square$

## B.3 Proof of Theorem 5.5

With this assumption, we come to present our main results.

**Theorem B.3.** *Under Assumptions 4.2 and 5.4, suppose $\epsilon < 1 - \gamma^\star(x)$. Let $p_{LLM}(x)$ be defined as in Assumption 5.4. Then set*

$$\gamma = \gamma^\star(x), \, m = \frac{\log[\epsilon^{-1} p_{ref}^{-1}(\tau^\star(x))]}{\Delta(x)}, \, \bar{n} := \frac{m}{p_{LLM}(x)},$$

$$n = \text{poly} \log \left( \frac{1}{\epsilon} \right) \begin{cases} M_{LLM}^{x,\epsilon}, & \text{if } M_{LLM}^{x,\epsilon} \leq \bar{n} \\ M_{\tau^\star(x)}^{x,\epsilon} + \sqrt{\bar{n} \left( M_{LLM}^{x,\epsilon} - M_{\tau^\star(x)}^{x,\epsilon} \right)}, \, else \, . \end{cases}$$

*for $\pi^\star$, the output policy of Algorithm 2 is $4\epsilon$-optimal to $\pi^\star$.*

*Remark* B.4. The sample complexity of Algorithm 2 has two different regimes, dependent on $M_{\text{LLM}}^{x,\epsilon}$, which represents the necessary samples to use vanilla BoN to solve $x$, and a threshold $\bar{n}$. When $M_{\text{LLM}}^{x,\epsilon} \leq \bar{n}$, our Algorithm 2 performs the same as vanilla BoN, which suggests sequential TTC only outperforms parallel BoN when the problem instance $x$ is hard to be resolved. When $M_{\text{LLM}}^{x,\epsilon}$ is large, the sample complexity of Algorithm 2 falls into another regime. It can be seen that the sample complexity can be shaved to $\sqrt{\bar{n} M_{\text{LLM}}^{x,\epsilon}}$ when the theoretical lower bound $M_{\tau^\star(x)}^{x,\epsilon} \ll M_{\text{LLM}}^{x,\epsilon}$, which suggests that our Algorithm 2 could save a lot of sample complexity when there exists a "best" reference policy $\pi_{\text{ref}}^{\tau^\star(x)}(\cdot \mid x)$ that finds the optimal actions efficiently.

We first need the following lemma modified from Lemma C.3 in Hu et al. (2024):

**Lemma B.5.** *For any $i \in [n]$, let $k$ denote the length of $h_i$, i.e., the number of actions in $h_i$. Then we have*

$$\frac{\pi_{LLM}(a \mid h_i)}{\pi_{ref}^{\tau^\star(x)}(a \mid x)} > \frac{1}{1 + e^{-k\Delta}/p_{ref}(\tau^\star(x))}.$$

*Proof of Lemma B.5.* For simplicity, let $i_1 < \cdots < i_k$ denote the indices of $a_i$ that have been put into $h$. By the product factorization and Assumption 5.4,

$$\log \frac{\mathbb{P}(h_i \mid \tau)}{\mathbb{P}(h_i \mid \tau^\star(x))} = \sum_{j=1}^{k} \log \frac{\pi_{\text{ref}}^\tau(a_{i_j} \mid x)}{\pi_{\text{ref}}^{\tau^\star(x)}(a_{i_j} \mid x)} \leq \sum_{j=1}^{k}(-\Delta) = -k\Delta, \tag{B.11}$$

which yields the claim after exponentiation.

Next, by the assumption of the pretraining distribution $\mathbb{P}_\mathcal{D}$, the predictive policy of the LLM is the Bayes mixture

$$\pi_{\text{LLM}}(\cdot \mid h_i) = \sum_\tau \pi_{\text{ref}}^\tau(\cdot \mid x) \cdot w_i(\tau), \qquad w_i(\tau) := \frac{p_{\text{ref}}(\tau)\mathbb{P}(h_i \mid \tau)}{\sum_{\tau'} p_{\text{ref}}(\tau')\mathbb{P}(h_i \mid \tau')}. \tag{B.12}$$

Using the likelihood–ratio bound (B.11), for any $\tau \neq \tau^\star(x)$,

$$\frac{w_i(\tau)}{w_i(\tau^\star(x))} = \frac{\mathbb{P}(h_i \mid \tau)}{\mathbb{P}(h_i \mid \tau^\star(x))} \cdot \frac{p_{\text{ref}}(\tau)}{p_{\text{ref}}(\tau^\star(x))} \leq e^{-k\Delta} \cdot \frac{p_{\text{ref}}(\tau)}{p_{\text{ref}}(\tau^\star(x))}. \tag{B.13}$$

Summing over $\{\tau : \tau \neq \tau^\star(x)\}$, we have

$$\frac{1 - w_m(\tau^\star(x))}{w_m(\tau^\star(x))} \leq e^{-m\Delta} \cdot \frac{1 - p_{\text{ref}}(\tau^\star(x))}{p_{\text{ref}}(\tau)}.$$

Then it is easy to obtain that

$$w_i(\tau^\star(x)) \geq 1 - \frac{1 - p_{\text{ref}}(\tau^\star(x))}{1 + p_{\text{ref}}(\tau^\star(x))(e^{k\Delta} - 1)} > \frac{1}{1 + e^{-k\Delta}/p_{\text{ref}}(\tau^\star(x))} \tag{B.14}$$

Therefore, for any action $a$, we have

$$\frac{\pi_{\text{LLM}}(a \mid h_i)}{\pi_{\text{ref}}^{\tau^\star(x)}(a \mid x)} \geq w_i(\tau^\star(x)) \geq \frac{1}{1 + e^{-k\Delta}/p_{\text{ref}}(\tau^\star(x))}. \tag{B.15}$$

$\square$

Next we have our overall proof of Theorem B.3.

*Proof of Theorem B.3.* Denote $i^\star$ be the first index $i$ satisfying $|h_i| = m$. Here we allow $i^\star > n$, in which case $|h_i| < m$ for any $1 \leq i \leq n$. By the bound established in Lemma B.5 and the selection of $m = \log[\epsilon^{-1}p_{\text{ref}}^{-1}(\tau^\star(x))]/\Delta$, for any $i > i^\star$, we have

$$\frac{\pi_{\text{LLM}}(a \mid h_i)}{\pi_{\text{ref}}^{\tau^\star(x)}(a \mid x)} \geq \frac{1}{1 + e^{-m\Delta}/p_{\text{ref}}(\tau^\star(x))} > 1 - \epsilon. \tag{B.16}$$

Let $M_0 = M_{\pi^\star(\cdot|x), \pi_{\text{ref}}^{\tau^\star(x)}(\cdot|x)}^\epsilon$. Using the definition of $\mathcal{E}_M$-divergence, we have

$$\mathcal{E}_{M_0}\left(\pi^*(\cdot \mid x), \pi_{\text{LLM}}(\cdot \mid h_i)\right) = \sum_{a \in \mathcal{A}} \max\{0, \pi^\star(a \mid x) - M_{\pi^\star(\cdot|x), \pi_{\text{ref}}^{\tau^\star(x)}(\cdot|x)}^\epsilon \cdot \pi_{\text{LLM}}(a \mid h_i)\}$$

$$\leq \sum_{a \in \mathcal{A}} \max\left\{0, \pi^\star(a \mid x) - M_{\pi^\star(\cdot|x), \pi_{\text{ref}}^{\tau^\star(x)}(\cdot|x)}^\epsilon \cdot (1 - \epsilon) \cdot \pi_{\text{ref}}^{\tau^\star(x)}(a \mid x)\right\}$$

$$\leq (1 - \epsilon) \cdot \sum_{a \in \mathcal{A}} \max\left\{0, \pi^\star(a \mid x) - M_{\pi^\star(\cdot|x), \pi_{\text{ref}}^{\tau^\star(x)}(\cdot|x)}^\epsilon \cdot \pi_{\text{ref}}^{\tau^\star(x)}(a \mid x)\right\} + \epsilon \sum_{a \in \mathcal{A}} \pi^\star(a \mid x)$$

$$\leq \mathcal{E}_{M_0}\left(\pi^*(\cdot \mid x), \pi_{\text{ref}}^{\tau^\star(x)}(\cdot \mid x)\right) + \epsilon$$

$$\leq 2\epsilon,$$

where the first inequality holds due to B.16. The second inequality holds due to $\max\{a, b + c\} \leq \max\{a, b\} + c$ for $a, b \in \mathbb{R}$ and $c > 0$. The third inequality holds due to $\mathcal{E}_{M_0}(\pi^*, \pi_{\text{ref}}^{\tau^*(x)}) > 0$. The last inequality holds because $M_0 = M^\epsilon_{\pi^*(\cdot|x), \pi_{\text{ref}}^{\tau^*(x)}(\cdot|x)}$ is the smallest $M$ such that $\mathcal{E}_M(\pi^*, \pi_{\text{ref}}^{\tau^*(x)}) \leq \epsilon$. Therefore, we have

$$M^{2\epsilon}_{\pi^*(\cdot|x), \pi_{\text{LLM}}(\cdot|h_i)} \leq M^\epsilon_{\pi^*(\cdot|x), \pi_{\text{ref}}^{\tau^*(x)}(\cdot|x)}.$$

Combined with the result of Theorem 4.6, we have for any $i > i^*$

$$M^{2\epsilon}_{\pi^*(\cdot|x), \pi_{\text{LLM}}(\cdot|h_i)} \leq M^\epsilon_{\pi^*(\cdot|x), \pi_{\text{ref}}^{\tau^*(x)}(\cdot|x)} \leq M^\epsilon_{\pi^*(\cdot|x), \pi_{\text{LLM}}(\cdot|x)}. \tag{B.17}$$

Note that for each step $i \leq i^*$, the sampled action $a_i$ is accepted into into $h_i$ with probability $p_{\text{LLM}}$, defined as:

$$p_{\text{LLM}} := \mathbb{P}_{a \sim \pi_{\text{LLM}}(\cdot|x)}\big(r(a, x) \geq \gamma^*(x)\big).$$

Therefore, $i^*$ satisfies a negative binomial distribution $N(m, p_{\text{LLM}})$, and its tail function $\mathbb{P}(i^* > k)$ for any $k$ is equal to $\mathbb{P}(j^* < m)$, where $j^*$ satisfies a Binomial distribution $B(k, p_{\text{LLM}})$, which is

$$\mathbb{P}(i^* > k) = \mathbb{P}\big(B(k, p_{\text{LLM}}) < m\big) \leq \exp\left(-\frac{(m - kp_{\text{LLM}})^2}{kp_{\text{LLM}}(1 - p_{\text{LLM}})}\right), \tag{B.18}$$

where we use the fact that the Binomial distribution is sub-Gaussian. Therefore, by selecting $k = \bar{n} = p_{\text{LLM}}^{-1} \cdot (3m + \log(1/\epsilon))$, we have

$$\mathbb{P}(i^* > \bar{n}) < \epsilon. \tag{B.19}$$

Next, we analyze the original statement.

By Theorem 5.1, we have

$$\mathbb{E}_{a \sim \pi^*}[r(a, x)] - \mathbb{E}_{a \sim \pi_{\text{reward}}}[r(a, x)]$$
$$\leq 2\epsilon + \mathbb{E}_{a_1, \dots, a_n}\left[\exp\left(-\frac{n^2}{\sum_{i=1}^n M^{2\epsilon}_{\pi^*(\cdot|x), \pi_{\text{LLM}}(\cdot|h_i)}}\right)\right]$$
$$= 2\epsilon + \mathbb{E}_{a_1, \dots, a_n}\left[\exp\left(-\frac{n^2}{\sum_{i=1}^{i^*} M^{2\epsilon}_{\pi^*(\cdot|x), \pi_{\text{LLM}}(\cdot|h_i)} + \sum_{i=i^*+1}^n M^{2\epsilon}_{\pi^*(\cdot|x), \pi_{\text{LLM}}(\cdot|h_i)}}\right)\right] \tag{B.20}$$
$$\leq 2\epsilon + \mathbb{E}_{a_1, \dots, a_n}\left[\exp\left(-\frac{n^2}{\min\{i^*, n\} M^{2\epsilon}_{\pi^*(\cdot|x), \pi_{\text{LLM}}(\cdot|x)} + (n - \min\{i^*, n\}) M^\epsilon_{\pi^*(\cdot|x), \pi_{\text{ref}}^{\tau^*(x)}(\cdot|x)}}\right)\right]$$

where the last line holds due to (B.17).

Next, we further decompose it based on the value of $i^*$:

$$\mathbb{E}_{a \sim \pi^*}[r(a, x)] - \mathbb{E}_{a \sim \pi_{\text{reward}}}[r(a, x)]$$
$$\leq 2\epsilon + \sum_{i=1}^\infty \mathbb{P}(i^* = i)\left[\exp\left(-\frac{n^2}{\min\{i, n\} M^{2\epsilon}_{\pi^*(\cdot|x), \pi_{\text{LLM}}(\cdot|x)} + (n - \min\{i, n\}) \cdot M^\epsilon_{\pi^*(\cdot|x), \pi_{\text{ref}}^{\tau^*(x)}(\cdot|x)}}\right)\right]$$
$$= 2\epsilon + \left(\sum_{i=1}^{\bar{n}} + \sum_{i=\bar{n}+1}^\infty\right) \mathbb{P}(i^* = i)$$
$$\cdot \left[\exp\left(-\frac{n^2}{\min\{i, n\} M^{2\epsilon}_{\pi^*(\cdot|x), \pi_{\text{LLM}}(\cdot|x)} + (n - \min\{i, n\}) \cdot M^\epsilon_{\pi^*(\cdot|x), \pi_{\text{ref}}^{\tau^*(x)}(\cdot|x)}}\right)\right]$$
$$\leq 2\epsilon$$
$$+ \sum_{i=1}^{\bar{n}} \mathbb{P}(i^* = i)\left[\exp\left(-\frac{n^2}{\min\{i, n\} M^{2\epsilon}_{\pi^*(\cdot|x), \pi_{\text{LLM}}(\cdot|x)} + (n - \min\{i, n\}) \cdot M^\epsilon_{\pi^*(\cdot|x), \pi_{\text{ref}}^{\tau^*(x)}(\cdot|x)}}\right)\right]$$

$$+ \mathbb{P}(i^\star > \bar{n})$$

$$\leq 2\epsilon$$

$$+ \sum_{i=1}^{\bar{n}} \mathbb{P}(i^\star = i)\left[ \exp\left( -\frac{n^2}{\min\{i,n\}M^\epsilon_{\pi^\star(\cdot|x),\pi_{\mathrm{LLM}}(\cdot|x)} + (n - \min\{i,n\})\cdot M^\epsilon_{\pi^\star(\cdot|x),\pi^{\tau^\star(x)}_{\mathrm{ref}}(\cdot|x)}} \right) \right]$$

$$+ \mathbb{P}(i^\star > \bar{n})$$

$$\leq 3\epsilon + \exp\left( -\frac{n^2}{\bar{n}M^\epsilon_{\pi^\star(\cdot|x),\pi_{\mathrm{LLM}}(\cdot|x)} + (n - \bar{n})\cdot M^\epsilon_{\pi^\star(\cdot|x),\pi^{\tau^\star(x)}_{\mathrm{ref}}(\cdot|x)}} \right) \tag{B.21}$$

where the second inequality holds because the exponential term is less than 1. The third inequality holds due to $M^{2\epsilon}_{\pi^\star(\cdot|x),\pi_{\mathrm{LLM}}(\cdot|x)} \leq M^\epsilon_{\pi^\star(\cdot|x),\pi_{\mathrm{LLM}}(\cdot|x)}$. The fourth inequality holds due to (B.19) and $\min\{i,n\} \leq \bar{n}$, $M^\epsilon_{\pi^\star(\cdot|x),\pi_{\mathrm{LLM}}(\cdot|x)} \geq M^\epsilon_{\pi^\star(\cdot|x),\pi^{\tau^\star(x)}_{\mathrm{ref}}(\cdot|x)}$.

When $M^\epsilon_{\pi^\star(\cdot|x),\pi_{\mathrm{LLM}}(\cdot|x)} \leq \bar{n}$, as $M^\epsilon_{\pi^\star(\cdot|x),\pi_{\mathrm{LLM}}(\cdot|x)} \geq M^\epsilon_{\pi^\star(\cdot|x),\pi^{\tau^\star(x)}_{\mathrm{ref}}(\cdot|x)}$, we have

$$\exp\left( -\frac{n^2}{\bar{n}M^\epsilon_{\pi^\star(\cdot|x),\pi_{\mathrm{LLM}}(\cdot|x)} + (n-\bar{n})\cdot M^\epsilon_{\pi^\star(\cdot|x),\pi^{\tau^\star(x)}_{\mathrm{ref}}(\cdot|x)}} \right)$$

$$\leq \exp\left( -\frac{n}{M^\epsilon_{\pi^\star(\cdot|x),\pi_{\mathrm{LLM}}(\cdot|x)}} \right).$$

Thus, it suffices to select $n = \mathrm{poly}\log(1/\epsilon)\cdot M^\epsilon_{\pi^\star(\cdot|x),\pi_{\mathrm{LLM}}(\cdot|x)}$.

Otherwise, using basic inequalities, we have

$$\sqrt{\log^2(1/\epsilon)\left[ M^\epsilon_{\pi^\star(\cdot|x),\pi^{\tau^\star(x)}_{\mathrm{ref}}(\cdot|x)} \right]^2 + 4\log(1/\epsilon)\bar{n}\left[ M^\epsilon_{\pi^\star(\cdot|x),\pi_{\mathrm{LLM}}(\cdot|x)} - M^\epsilon_{\pi^\star(\cdot|x),\pi^{\tau^\star(x)}_{\mathrm{ref}}(\cdot|x)} \right]}$$

$$\leq \log(1/\epsilon)M^\epsilon_{\pi^\star(\cdot|x),\pi^{\tau^\star(x)}_{\mathrm{ref}}(\cdot|x)} + 2\sqrt{\log(1/\epsilon)}\cdot\sqrt{\bar{n}\left[ M^\epsilon_{\pi^\star(\cdot|x),\pi_{\mathrm{LLM}}(\cdot|x)} - M^\epsilon_{\pi^\star(\cdot|x),\pi^{\tau^\star(x)}_{\mathrm{ref}}(\cdot|x)} \right]},$$

where we use the inequality $\sqrt{a+b} \leq \sqrt{a} + \sqrt{b}$ for $a,b > 0$. Therefore, when

$$n = \log(1/\epsilon)M^\epsilon_{\pi^\star(\cdot|x),\pi^{\tau^\star(x)}_{\mathrm{ref}}(\cdot|x)} + 2\sqrt{\log(1/\epsilon)}\cdot\sqrt{\bar{n}\left[ M^\epsilon_{\pi^\star(\cdot|x),\pi_{\mathrm{LLM}}(\cdot|x)} - M^\epsilon_{\pi^\star(\cdot|x),\pi^{\tau^\star(x)}_{\mathrm{ref}}(\cdot|x)} \right]},$$

we have

$$n^2 \geq \log(1/\epsilon)n + \log(1/\epsilon)\bar{n}\left[ M^\epsilon_{\pi^\star(\cdot|x),\pi_{\mathrm{LLM}}(\cdot|x)} - M^\epsilon_{\pi^\star(\cdot|x),\pi^{\tau^\star(x)}_{\mathrm{ref}}(\cdot|x)} \right].$$

Thus, $\mathbb{E}_{a\sim\pi^\star}[r(a,x)] - \mathbb{E}_{a\sim\pi_{\mathrm{reward}}}[r(a,x)] \leq 4\epsilon$. It suffices to select

$$n = \mathrm{poly}\log(1/\epsilon)\left[ M^{x,\epsilon}_{\tau^\star(x)} + \sqrt{\bar{n}\left( M^{x,\epsilon}_{\mathrm{LLM}} - M^{x,\epsilon}_{\tau^\star(x)} \right)} \right].$$

$\square$

Next we prove Theorem 5.5. We show the formal version of Theorem 5.5 here, which includes an upper bound of $\epsilon$ derived from Theorem B.3.

**Theorem B.6** (Formal Version of Theorem 5.5). *Following the hyperparameter selection in Theorem B.3, denote $d(x) := (1 + e^{-\Delta(x)}/p_{ref}(\tau^\star(x)))$. Then, for sufficiently small $\epsilon > 0$ satisfying*

$$\epsilon \leq \min\left\{ 1 - \gamma^\star(x), \frac{C^\star_{LLM}(x)\mathbb{P}_{a\sim\pi^{\tau^\star(x)}_{ref}(\cdot|x)}(r(a,x) \geq \gamma^\star(x))\Delta(x)}{\log p^{-1}_{ref}(\tau^\star(x))}\cdot\frac{\kappa(x)}{d^2(x)} \right\},$$

*Algorithm 2 finds an $\epsilon$-optimal policy to $\pi^\star$ with sample complexity*

$$n = \mathrm{poly}\log(1/\epsilon)\cdot\kappa(x)\cdot\frac{C^\star_{LLM}(x)}{\epsilon}.$$

*Proof of Theorem B.6.* Using Lemma B.5, we have

$$\frac{\pi_{\text{LLM}}(a \mid h_i)}{\pi_{\text{ref}}(a, x; \tau^\star(x))} > \frac{1}{1 + e^{-k\Delta(x)}/p_{\text{ref}}(\tau^\star(x))}, \tag{B.22}$$

where $k$ is the length of $h_i$. Note that

$$
\begin{aligned}
&C(\pi^\star(\cdot, x), \pi_{\text{ref}}(\cdot, x; \tau^\star(x))) \\
&= \mathbb{E}_{a \sim \pi^\star(\cdot, x)} \frac{\pi^\star(a, x)}{\pi_{\text{ref}}(a, x; \tau^\star(x))} \\
&= \mathbb{E}_{a \sim \pi^\star(\cdot, x)} \frac{\pi^\star(a, x)}{\pi_{\text{LLM}}(a \mid x)} \cdot \frac{\pi_{\text{LLM}}(a \mid x)}{\pi_{\text{ref}}(a, x; \tau^\star(x))} \\
&= \mathbb{E}_{a \sim \pi^\star(\cdot, x)} \frac{\pi^\star(a, x)}{\pi_{\text{LLM}}(a \mid x)} \cdot \frac{\sum_{\tau \in \mathcal{T}_{\text{ref}}} p_{\text{ref}}(\tau) \pi_{\text{ref}}(a, x; \tau)}{\pi_{\text{ref}}(a, x; \tau^\star(x))} \\
&\leq \mathbb{E}_{a \sim \pi^\star(\cdot, x)} \frac{\pi^\star(a, x)}{\pi_{\text{LLM}}(a \mid x)} \cdot \frac{\sum_{\tau \neq \tau^\star(x)} p_{\text{ref}}(\tau) e^{-\Delta(x)} \pi_{\text{ref}}(a, x; \tau^\star(x)) + p_{\text{ref}}(\tau^\star(x)) \pi_{\text{ref}}(a, x; \tau^\star(x))}{\pi_{\text{ref}}(a, x; \tau^\star(x))} \\
&\leq \min\{e^{-\Delta(x)}, p_{\text{ref}}(\tau^\star(x))\} \cdot C(\pi^\star(\cdot, x), \pi_{\text{LLM}}(\cdot \mid x)), \tag{B.23}
\end{aligned}
$$

where the first inequality holds due to Assumption 5.4 and $\text{supp}(\pi^\star(\cdot, x)) \subseteq \{a \ : \ \log \pi_{\text{ref}}(a, x; \tau^\star(x)) - \sup_{\tau \neq \tau^\star(x)} \log \pi_{\text{ref}}(a, x; \tau) \geq \Delta(x)\}$, the last one holds by calculation. Therefore, by selecting $\epsilon$ to be small enough, we have our statement by (B.21). With similar argument, we have

$$
\begin{aligned}
&C(\pi^\star(\cdot, x), \pi_{\text{LLM}}(a \mid h_i)) \\
&\leq (1 + e^{-k\Delta(x)}/p_{\text{ref}}(\tau^\star(x))) C(\pi^\star(\cdot, x), \pi_{\text{ref}}(\cdot, x; \tau^\star(x))) \\
&\leq \underbrace{(1 + e^{-\Delta(x)}/p_{\text{ref}}(\tau^\star(x)))}_{d(x)} C(\pi^\star(\cdot, x), \pi_{\text{ref}}(\cdot, x; \tau^\star(x))) \\
&\leq d(x) \cdot \kappa(x) \cdot C(\pi^\star(\cdot, x), \pi_{\text{LLM}}(\cdot \mid x)),
\end{aligned}
$$

where the first inequality holds due to (B.22), the last one holds due to (B.23).

We still denote $i^\star$ be the first index satisfying $|h_i| = m$. Then $i^\star$ is a random variable as follows: it is a summation of a series of Geometric distribution. Then we have

$$i^\star = G(p_1) + \cdots + G(p_m), p_i = \mathbb{P}_{a \sim \pi_{\text{LLM}}(\cdot \mid h_i)}(r(a, x) \geq \gamma^\star(x)). \tag{B.24}$$

We know that each $G(p_i)$ is an $(1/p_i^2, 1/p_i)$-sub-exponential distribution. Using Bernstein inequality, we know that with probability at least $1 - \epsilon$, we have

$$
\begin{aligned}
i^\star &\leq 2\bigg( \sum_{i=1}^m \frac{1}{p_i} + \sqrt{\sum_{i=1}^m \frac{1}{p_i^2} \log \epsilon^{-1}} + \frac{\log \epsilon^{-1}}{\min\{p_i\}} \bigg) \\
&\leq \underbrace{\frac{4md(x) \log \epsilon^{-1}}{\mathbb{P}_{a \sim \pi_{\text{ref}}(\cdot, x; \tau^\star(x))}(r(a, x) \geq \gamma^\star(x))}}_{\bar{n}}, \tag{B.25}
\end{aligned}
$$

where we use the fact that

$$
\begin{aligned}
p_i &= \mathbb{P}_{a \sim \pi_{\text{LLM}}(\cdot \mid h_i)}(r(a, x) \geq \gamma^\star(x)) \\
&= \mathbb{E}_{a \sim \pi_{\text{ref}}(\cdot, x; \tau^\star(x))} \frac{\pi_{\text{LLM}}(a \mid h_i)}{\pi_{\text{ref}}(a, x; \tau^\star(x))} \mathbb{1}(r(a, x) \geq \gamma^\star(x)) \\
&\geq \frac{1}{d(x)} \mathbb{P}_{a \sim \pi_{\text{ref}}(\cdot, x; \tau^\star(x))}(r(a, x) \geq \gamma^\star(x)).
\end{aligned}
$$

Hence, selecting $\bar{n}$ as (B.25) suggests, following the same step from (B.20), we have

$$
\begin{aligned}
& \mathbb{E}_{a\sim\pi^\star}[r(a,x)] - \mathbb{E}_{a\sim\pi_{\text{reward}}}[r(a,x)] \\
& \leq \epsilon + \mathbb{E}_{a_1,\dots,a_n}\left[\exp\left(-\frac{n^2}{\sum_{i=1}^{i^\star} M^\epsilon_{\pi^\star(\cdot,x),\pi_{\text{LLM}}(\cdot|h_i)} + \sum_{i=i^\star+1}^{n} M^\epsilon_{\pi^\star(\cdot,x),\pi_{\text{LLM}}(\cdot|h_i)}}\right)\right] \\
& \leq \epsilon + \mathbb{E}_{a_1,\dots,a_n}\left[\exp\left(-\frac{n^2\epsilon}{\sum_{i=1}^{i^\star} C(\pi^\star(\cdot,x),\pi_{\text{LLM}}(\cdot\mid h_i)) + \sum_{i=i^\star+1}^{n} C(\pi^\star(\cdot,x),\pi_{\text{LLM}}(\cdot\mid h_i))}\right)\right] \\
& \leq \epsilon + \mathbb{E}_{a_1,\dots,a_n}\left[\exp\left(-\frac{\frac{n^2\epsilon}{\kappa(x)\cdot C(\pi^\star(\cdot,x),\pi_{\text{LLM}}(\cdot|x))}}{\sum_{i=1}^{i^\star} d(x) + \sum_{i=i^\star+1}^{n} 1}\right)\right] \\
& \leq \epsilon + \mathbb{E}_{a_1,\dots,a_n}\left[\exp\left(-\frac{\frac{n^2\epsilon}{\kappa(x)\cdot C(\pi^\star(\cdot,x),\pi_{\text{LLM}}(\cdot|x))}}{\min\{\bar{n},n\}d(x) + (n - \min\{\bar{n},n\})}\right)\right]
\end{aligned}
\tag{B.26}
$$

Finally, selecting

$$
\epsilon \leq \min\left\{1-\gamma^\star(x),\ \frac{C^\star_{\text{LLM}}(x)\mathbb{P}_{a\sim\pi_{\text{ref}}(\cdot,x;\tau^\star(x))}(r(a,x)\geq\gamma^\star(x))\Delta(x)}{\log p_{\text{ref}}^{-1}(\tau^\star(x))}\cdot\frac{\kappa(x)}{d^2(x)}\right\},
$$

satisfies (B.26) to be $2\epsilon$-optimal. $\qquad\square$

## C  RF-SeqBoN with Burn-In

From a theoretical perspective, we introduce another variant of RF-SeqBoN with an explicit burn-in phase (Algorithm 4). We continue to use a reward filter to collect responses; however, in contrast to Algorithm 2, we update the input only after a sufficiently long sequence has been collected. This design avoids the detrimental effect of the early stage, where the limited collected response may be insufficiently good. The theoretical guarantee of Algorithm 4 is presented as follows:

---

**Algorithm 4** RF-SeqBoN with Burn-In

**Require:** $\pi_{\text{LLM}}, x, r(\cdot,\cdot), \gamma, m$
 1: Initialize $\bar{h} \leftarrow \langle x\rangle$
 2: **for** $i = 1,\dots,n$ **do**
 3:     **If** $|\bar{h}| \geq m$ **then** $h_i \leftarrow \bar{h}$ **else** $h_i \leftarrow \langle x\rangle$
 4:     Sample $a_i \sim \pi_{\text{LLM}}(\cdot\mid h_i)$
 5:     **If** $r(a_i,x) \geq \gamma$ **then** $\bar{h} \leftarrow \bar{h}\parallel a_i$
 6: **end for**
**Ensure:** $\widehat{a} = \arg\max_{i\in[n]} r(a_i,x)$

---

**Corollary C.1.** *Following the assumption on $\pi^\star$ defined in Theorem B.3, denote $C^\star_{LLM}(x)$ as in Proposition 4.5 and denote $\kappa(x) := \max\left\{p_{ref}(\tau^\star(x)),\ e^{-\Delta(x)}\right\}$. Then, for sufficiently small $\epsilon > 0$ satisfying*

$$
\epsilon \leq \min\left\{1-\gamma^\star(x),\ \frac{C^\star_{LLM}(x)\,p_{LLM}(x)\,\Delta(x)}{\log p_{ref}^{-1}(\tau^\star(x))},\ \frac{C^\star_{LLM}(x)\,p_{LLM}(x)\,\Delta(x)}{\log p_{ref}^{-1}(\tau^\star(x))}\cdot\frac{\kappa^2(x)}{1-\kappa(x)}\right\},
$$

*Algorithm 4 finds an $\epsilon$-optimal policy to $\pi^\star$ with sample complexity*

$$
n = \text{poly}\log(1/\epsilon)\cdot\kappa(x)\cdot\frac{C^\star_{LLM}(x)}{\epsilon}.
$$

Compared with Theorem B.6, the condition on $\epsilon$ is less stringent, and thus the result applies to a more general setting. However, the waiting time required to collect a sufficient number of high-reward responses may be prohibitively long in practice, and therefore, this variant may not be well-suited for practical use. Nevertheless, we include this result primarily for its theoretical value, which we believe may be inspiring for future research.

*Remark* C.2. Corollary C.1 and Theorem B.6 also reveal that $\epsilon$ must be sufficiently small. The admissible range of $\epsilon$ is controlled jointly by the gap $\Delta(x)$ and the probability $p_{\mathrm{LLM}}(x)$ that the LLM outputs actions with reward above the threshold. These findings are consistent with empirical observations: sequential TTC methods tend to outperform parallel TTC methods when the optimal trajectory is relatively rare but significantly more rewarding than alternatives (Snell et al., 2024).

*Proof of Corollary C.1.* We follow the proof of Theorem B.3 from (B.21). By Definition 4.4, using the fact that

$$M^\epsilon_{\pi^\star(\cdot,x),\pi_{\mathrm{LLM}}(\cdot|x)} \leq \frac{C(\pi^\star(\cdot,x),\pi_{\mathrm{LLM}}(\cdot\mid x))}{\epsilon}, \; M^\epsilon_{\pi^\star(\cdot,x),\pi_{\mathrm{ref}}(\cdot,x;\tau^\star(x))} \leq \frac{C(\pi^\star(\cdot,x),\pi_{\mathrm{ref}}(\cdot,x;\tau^\star(x)))}{\epsilon},$$

Next, we bound $C(\pi^\star(\cdot,x),\pi_{\mathrm{ref}}(\cdot,x;\tau^\star(x)))$. Note that

$$
\begin{aligned}
&C(\pi^\star(\cdot,x),\pi_{\mathrm{ref}}(\cdot,x;\tau^\star(x))) \\
&= \mathbb{E}_{a\sim\pi^\star(\cdot,x)} \frac{\pi^\star(a,x)}{\pi_{\mathrm{ref}}(a,x;\tau^\star(x))} \\
&= \mathbb{E}_{a\sim\pi^\star(\cdot,x)} \frac{\pi^\star(a,x)}{\pi_{\mathrm{LLM}}(a\mid x)} \cdot \frac{\pi_{\mathrm{LLM}}(a\mid x)}{\pi_{\mathrm{ref}}(a,x;\tau^\star(x))} \\
&= \mathbb{E}_{a\sim\pi^\star(\cdot,x)} \frac{\pi^\star(a,x)}{\pi_{\mathrm{LLM}}(a\mid x)} \cdot \frac{\sum_{\tau\in\mathcal{T}_{\mathrm{ref}}} p_{\mathrm{ref}}(\tau)\pi_{\mathrm{ref}}(a,x;\tau)}{\pi_{\mathrm{ref}}(a,x;\tau^\star(x))} \\
&\leq \mathbb{E}_{a\sim\pi^\star(\cdot,x)} \frac{\pi^\star(a,x)}{\pi_{\mathrm{LLM}}(a\mid x)} \cdot \frac{\sum_{\tau\neq\tau^\star(x)} p_{\mathrm{ref}}(\tau)e^{-\Delta(x)}\pi_{\mathrm{ref}}(a,x;\tau^\star(x)) + p_{\mathrm{ref}}(\tau^\star(x))\pi_{\mathrm{ref}}(a,x;\tau^\star(x))}{\pi_{\mathrm{ref}}(a,x;\tau^\star(x))} \\
&\leq \min\{e^{-\Delta(x)}, p_{\mathrm{ref}}(\tau^\star(x))\} \cdot C(\pi^\star(\cdot,x),\pi_{\mathrm{LLM}}(\cdot\mid x)), \hspace{3cm} \text{(C.1)}
\end{aligned}
$$

where the first inequality holds due to Assumption 5.4 and $\mathrm{supp}(\pi^\star(\cdot,x)) \subseteq \{a : \log\pi_{\mathrm{ref}}(a,x;\tau^\star(x)) - \sup_{\tau\neq\tau^\star(x)} \log\pi_{\mathrm{ref}}(a,x;\tau) \geq \Delta(x)\}$, the last one holds by calculation. Therefore, by selecting $\epsilon$ to be small enough, we have our statement using a similar argument to (B.21).

$\square$

# D Experiments

## D.1 Hyperparameters and Dataset-Specific Settings

Unless otherwise specified, decoding settings are identical across methods and datasets. Table 1 lists the global settings held fixed; Table 2 provides per-dataset limits (maximum output tokens and maximum context length) and the reward-filter threshold $\gamma$ used with the PRM Llama3.1-8B-PRM-Deepseek-Data in the main results (Figure 1). Token budgets are adjusted by dataset to accommodate problem difficulty and solution length.

*Table 1.* Key hyperparameters of experiments across all configurations.

| Flag / Name | Value | Description |
|---|---|---|
| --history_budget | 3 | Max number of recent solutions retained in history. |
| --temperature | 0.8 | Sampling temperature of the foundation model. |
| --top_p | 0.95 | Nucleus sampling parameter. |
| --prm_agg | last-step | PRM aggregation strategy over reasoning steps. |

*Table 2.* Dataset specific parameters used in main results (Figure 1).

| Dataset | Max output tokens | Max context length | Reward- filter threshold $\gamma$ |
|---|---|---|---|
| MATH500 | 2048 | 8192 | 0.97 |
| GPQA-Diamond | 4096 | 16384 | 0.92 |
| AMC'23 | 8192 | 32768 | 0.92 |
| AIME'24 | 8192 | 32768 | 0.95 |
| AIME'25 | 8192 | 32768 | 0.90 |

### D.2 Prompt Templates

We use a unified chat-style prompt across all methods and datasets. Each query is formatted as a sequence of (`system`, `user`, `assistant`) messages: a dataset-specific *system* instruction, the *user* problem text, and (optionally) a short *assistant* history of previously accepted solutions.

**Message construction (used by PureSeq and RF-SeqBoN).** When history is present, we add a brief self-critique instruction before requesting a fresh, complete solution and a single final-answer line. The high-level template is:

```
messages = [
  {"role": "system", "content": SYSTEM_PROMPT(dataset)},
  {"role": "user",   "content": PROBLEM_TEXT},

  # If history exists, include up to # history_budget most recent
  # accepted solutions:
  {"role": "assistant", "content": ACCEPTED_{-k}}, {"role": "user",
  "content": "Try again with a different approach:"},
  ... (repeat for up to # history_budget earlier accepted solutions) ...
  {"role": "assistant", "content": ACCEPTED_{-1}},
  {"role": "user", "content":
    "The previous solution(s) may contain errors.
     Before solving, briefly critique the previous attempt(s)
     in 2 to 3 bullet points.
     Then provide a COMPLETE and CONCISE corrected solution
     from scratch that addresses those issues.
     End with exactly one line containing the final answer:"}
]
```

**Dataset-specific `system_prompt`.**

- **MATH500.** Concise for simple items; step-by-step sections for harder ones; end with "$\boxed{\text{answer}}$".

- **GPQA- Diamond.** Expert scientific reasoning, step-by-step elimination, and final line: "*The answer is (X)*" where $X \in \{A, B, C, D\}$.

- **AIME'24.** Expert mathematician; write math steps only; final boxed integer $\boxed{000\text{--}999}$.

**Notes about dialog prompts.**

- PureSeq and RF-SeqBoN share the same prompt template and critique instructions; RF-SeqBoN differs only in the reward-filtered acceptance and history management.

- We cap the history window at the number of `history_budget` recent accepted solutions.

- The final-answer line is enforced to simplify exact-match evaluation.

### D.3 Additional Ablation Studies

#### D.3.1 CHOICE OF PROMPT TEMPLATE

To address the potential concern that self-correction or reflection type of prompts may make sequential BoN win over Naive BoN, we conducted the ablation study on all three algorithms with the similar template below (PureSeq and RF-SeqBoN have the blue colored text to incorporate previous candidate solutions, which BoN doesn't, everything else remains the same) on Qwen3–4B–Instruct model and MATH500 benchmark dataset. Results are shown in Figure 4. We observe that RF-SeqBoN consistently outperforms both BoN and PureSeq as the generation budget $N$ increases under the updated prompt. Moreover, the results obtained with the updated prompt template

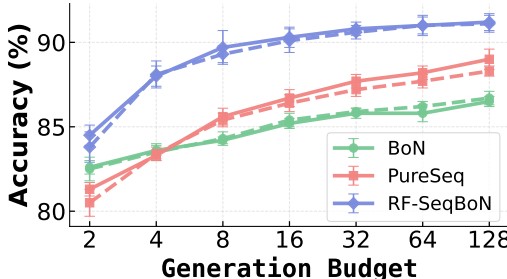

*Figure 4.* Ablation study of prompt template on MATH500 with Qwen3-4B-Instruct foundation model. Solid lines correspond to the prompt in the main study, while dashed lines indicate the updated prompt. The points and error bars show the mean and standard deviation from five repeated experiments, respectively.

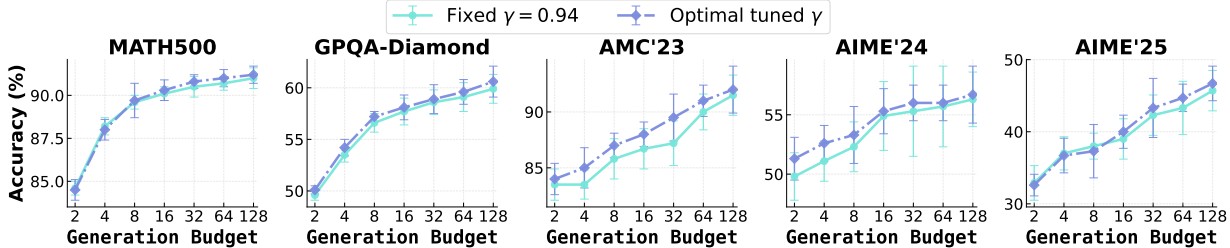

*Figure 5.* Ablation study comparing a fixed reward threshold $\gamma = 0.94$ against the dataset-specific optimal tuned threshold across five benchmarks. All experiments use Qwen3-4B-Instruct as the foundation model. Points denote the mean performance, with error bars representing the standard deviation over five repeated experiments.

(dashed lines) closely match those reported in the main study (solid lines), indicating that the performance of our method is not sensitive to the choice of prompt template.

---

**Prompt:** You are a careful problem-solving assistant for challenging math and reasoning problems.

[Problem] PROBLEM_TEXT

(PureSeq and RF-SeqBoN only:) [Previous candidate solutions] PREVIOUS_SOLUTIONS

The previous candidate solutions above are provided only as extra context reference.

Your task:
- Solve the problem from scratch.
- Write a single, clear, self-contained, correct and concise solution.
- Show your reasoning step by step.
- End with exactly one line containing the final answer: ⎡answer⎤.

---

### D.3.2 SENSITIVITY TO THE REWARD THRESHOLD

A natural question is whether RF-SeqBoN requires dataset-specific tuning of the reward threshold $\gamma$. As discussed in the main ablation on the choice of $\gamma$, the best-performing threshold does vary across datasets. However, this variation is relatively limited: the optimal values fall within a narrow range, approximately from $0.90$ to $0.97$. This suggests that while $\gamma$ controls an important quality-filtering tradeoff, RF-SeqBoN is not overly sensitive to fine-grained dataset-specific threshold tuning.

To further evaluate this point, we compare the dataset-specific optimal threshold with a single fixed threshold $\gamma = 0.94$ across all five benchmarks. The results are shown in Figure 5. Overall, the fixed threshold remains consistently competitive with the tuned threshold across MATH500, GPQA-Diamond, AMC'23, AIME'24, and AIME'25. Although the tuned threshold can yield modest gains on some datasets and budgets, the performance gap is generally small, and the fixed-threshold variant preserves the same increasing trend as the generation budget $N$ grows.

These results indicate that RF-SeqBoN does not rely on delicate per-dataset tuning of $\gamma$ to achieve strong performance. In practical deployment settings where the task distribution may be unknown or where validation-based tuning is inconvenient, a shared threshold such as $\gamma = 0.94$ can still provide a robust default choice. This supports the interpretation that the main benefit of RF-SeqBoN comes from its reward-filtered history construction mechanism rather than from carefully overfitting the threshold to each benchmark.

### D.4 Computation-Time Comparison and Discussion

#### D.4.1 MATCHED COMPUTATION-TIME BUDGET COMPARISON

We evaluate BoN, PureSeq, and RF-SeqBoN under a matched computation-time budget and summarize the results in Figure 6. Since BoN terminates the earliest, we stop all methods at the time when BoN finishes to ensure a fair comparison.

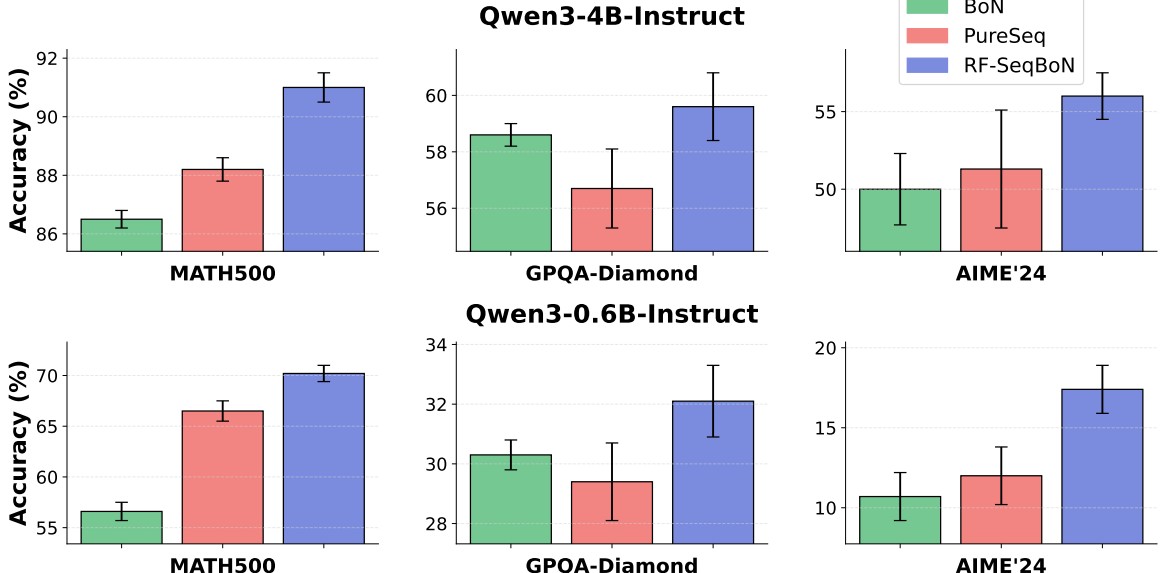

*Figure 6.* Accuracy comparison for BoN, PureSeq and RF-SeqBoN under the same test-time budget. The bar heights and error bars show the mean and standard deviation from five repeated experiments, respectively. The RF-SeqBoN still dominates the other two in all settings.

Due to the inherent uncertainty in API usage latency, we do not include results for GPT5-Nano, the closed-weight backbone model, in this comparison. As shown in Figure 6, RF-SeqBoN consistently outperforms both PureSeq and BoN across all benchmarks and backbone models, further supporting our claim regarding its computational efficiency. In addition, the accuracy trends in Figure 6 closely mirror those in Figure 1, indicating that comparisons based on generation budget are well aligned with those based on actual computation time.

### D.4.2 WALL-CLOCK LATENCY–ACCURACY TRADEOFF

We further examine whether the accuracy gains of RF-SeqBoN come at a reasonable wall-clock cost. Since RF-SeqBoN performs sequential generation and reward scoring, it is not expected to match the latency of fully parallel BoN. However, in our implementation, the sequential context growth is explicitly bounded by setting `history_budget = 3`, so at most three previous candidate solutions are kept in the prompt.

To evaluate the practical tradeoff, we measure the average first-token latency for a single query at generation budget $N = 128$ using a single NVIDIA A6000 GPU and the Qwen3-4B-Instruct foundation model. Figure 7 reports the resulting latency–accuracy Pareto comparison on MATH500 Levels 3–5. As expected, BoN achieves the lowest latency. Nevertheless, RF-SeqBoN provides a more favorable tradeoff than PureSeq, achieving consistently higher accuracy under comparable or lower latency. Moreover, on the more challenging Level 4–5 subsets, RF-SeqBoN attains noticeably higher accuracy than BoN with only a moderate increase in latency. These results suggest that RF-SeqBoN should not be viewed as a latency-minimizing method, but rather as a sequential test-time compute strategy that offers a favorable Pareto tradeoff between wall-clock cost and reasoning accuracy under bounded context-growth overhead.

### D.5 Additional Statistics on Filtered Answers for Different $\gamma$

As a complement to Section 6.1 (Choice of hyperparameter $\gamma$), Table 3 reports how many filtered answers remain in the LLM context under

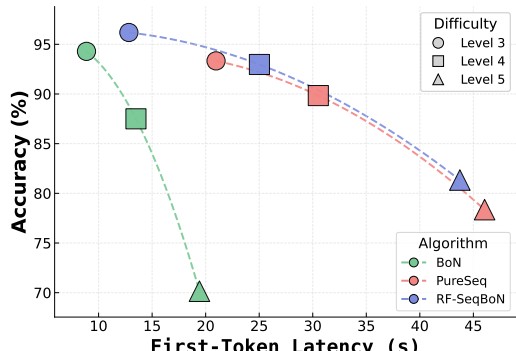

*Figure 7.* Wall-clock latency–accuracy tradeoff on MATH500 Levels 3–5 at generation budget $N = 128$. The x-axis reports the average first-token latency wall-clock time (in second) for a single query measured on one NVIDIA A6000 GPU, while the y-axis reports accuracy. RF-SeqBoN achieves a favorable Pareto tradeoff, especially on harder difficulty levels.

RF-SeqBoN when the generation budget is fixed at $N = 128$ and the history budget is set to `history_budget= 3` on MATH-500. We observe that the number of retained answers remains highly stable, ranging from 0.93 to 0.97 on average,

*Table 3.* Number of MATH500 questions that contribute $k = 3, 2, 1,$ or 0 filtered answers to the LLM context, under RF-SeqBoN when $N = 128$ and `-history_budget= 3`, for different values of threshold $\gamma$.

| $\gamma$ | # Questions with $k$ filtered answers | | | |
|---|---|---|---|---|
| | $k = 3$ | $k = 2$ | $k = 1$ | $k = 0$ |
| 0.90 | 418 | 27 | 39 | 16 |
| 0.93 | 399 | 44 | 38 | 19 |
| 0.95 | 391 | 46 | 39 | 24 |
| 0.97 | 383 | 50 | 41 | 26 |
| 0.99 | 332 | 75 | 64 | 29 |

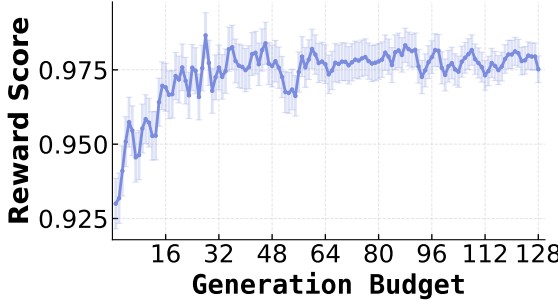

*Figure 8.* Reward score trend of RF-SeqBoN for the given problem. Points and error bars denote the mean and standard deviation across five repeated experiments.

which is consistent with the trends shown in Figure 3(a).

### D.6   Example of a Non-Monotonic Reward-Score Trend

We attach in Figure 8 the reward–score trajectory for a representative question from the MATH500 dataset (shown below), generated by the Qwen3-4B-Instruct foundation model and evaluated by the Llama3.1–8B–PRM–Deepseek–Data process reward model (aggregation taken at the final step), under the RF-SeqBoN algorithms.

The resulting curve shows that the reward-score sequence as a function of the generation budget is not strictly monotonically increasing, but instead exhibits an overall upward trend with small local fluctuations.

---

### # MATH500:

*Problem:*
You have seven bags of gold coins. Each bag has the same number of gold coins. One day, you find a bag of 53 coins. You decide to redistribute the number of coins you have so that all eight bags you hold have the same number of coins. You successfully manage to redistribute all the coins, and you also note that you have more than 200 coins. What is the smallest number of coins you could have had before finding the bag of 53 coins?

---

### D.7   Solution Examples

We present representative solutions generated by the Qwen3-4B-Instruct-2507 backbone across three benchmark datasets to illustrate the answer templates induced by the three algorithms. Text is color-coded as follows: brown for the dataset name, gray for the problem statement, and green, coral red, and purple for the solution segments produced by the BoN, PureSeq, and RF-SeqBoN algorithms, respectively (with colors matched to the plots throughout the paper).

---

### # MATH500:

*Problem:*
Tom got a Mr. Potato Head for his birthday. It came with 3 hairstyles, 2 sets of eyebrows, 1 pair of googly eyes, 2 sets of ears, and 2 sets of lips, a pair of regular shoes, and a bonus pair of clown shoes. If a complete Mr. Potato Head personality includes eyebrows, eyes, ears, lips, shoes and optionally hair, how many different wacky personalities can Tom come up with? Note that Mr. Potato Head can be bald.

Note: You cannot "mix and match". For example, you cannot take the left eyebrow from one pair and the right eyebrow from the other pair.

---

### BoN Solution Example (Incorrect)

**Solution:**

**Step 1.** Identify the independent choices for each required part of a complete Mr. Potato Head personality:

- Eyebrows: 2 different sets $\Rightarrow$ 2 choices.

- Eyes: 1 pair of googly eyes $\Rightarrow$ 1 choice.

- Ears: 2 different sets $\Rightarrow$ 2 choices.

- Lips: 2 different sets $\Rightarrow$ 2 choices.

- Shoes: 1 regular pair and 1 clown pair $\Rightarrow$ 2 choices.

- Hair: there are 3 hairstyles, and Mr. Potato Head can also be bald. We (incorrectly) treat this as giving 3 total options for hair (including the bald option).

**Step 2.** Since a personality consists of exactly one choice from each category (with hair being optional but already accounted for in the 3 options above), we multiply the number of choices:

$$\text{Total personalities} = 3 \times 2 \times 1 \times 2 \times 2 \times 2.$$

**Step 3.** Compute the product:

$$3 \times 2 = 6, \quad 6 \times 1 = 6, \quad 6 \times 2 = 12, \quad 12 \times 2 = 24, \quad 24 \times 2 = 48.$$

Therefore, the number of different wacky personalities Tom can come up with is

$$\boxed{48}$$

---

### PureSeq Solution Example (Incorrect)

**Solution:**

**Step 1.** As in previous attempts, we first list all the required components of a complete Mr. Potato Head personality and count the choices for each:

- Eyebrows: there are 2 distinct sets $\Rightarrow$ 2 choices.

- Eyes: there is 1 pair of googly eyes $\Rightarrow$ 1 choice.

- Ears: there are 2 distinct sets $\Rightarrow$ 2 choices.

- Lips: there are 2 distinct sets $\Rightarrow$ 2 choices.

- Shoes: there is 1 pair of regular shoes and 1 pair of clown shoes $\Rightarrow$ 2 choices.

- Hair: unlike the previous candidate, we now explicitly include the bald option along with the 3 hairstyles, so there are 3 hairstyles + bald = 4 choices for hair.

**Step 2.** A complete personality must include exactly one choice from each mandatory category (eyebrows, eyes, ears, lips, shoes), and either one of the 3 hairstyles or the bald option. Since these choices are independent, we multiply:

$$\text{Total personalities} = 4 \times 2 \times 1 \times 2 \times 2 \times 2.$$

**Step 3.** Compute the product step by step:

$$4 \times 2 = 8, \quad 8 \times 1 = 8, \quad 8 \times 2 = 16, \quad 16 \times 2 = 32, \quad 32 \times 2 = 60.$$

Therefore, the number of different wacky personalities Tom can come up with is

$$\boxed{60}$$

---

**RF-SeqBoN Example (Correct)**

**Solution:**

**Step 1.** Determine which parts are required and how many options there are for each. A complete personality must have:

- Eyebrows: 2 distinct sets $\Rightarrow$ 2 choices.

- Eyes: 1 pair of googly eyes $\Rightarrow$ 1 choice.

- Ears: 2 distinct sets $\Rightarrow$ 2 choices.

- Lips: 2 distinct sets $\Rightarrow$ 2 choices.

- Shoes: 1 regular pair and 1 clown pair $\Rightarrow$ 2 choices.

Hair is optional, but Tom has 3 hairstyles and the option to be bald, so:

$$\text{Hair choices} = 3 \text{ hairstyles} + 1 \text{ (bald)} = 4 \text{ choices.}$$

**Step 2.** First count the combinations of the mandatory parts (eyebrows, eyes, ears, lips, shoes). Since you cannot mix and match within a set, each set is a single atomic choice:

$$\text{Mandatory combinations} = 2 \times 1 \times 2 \times 2 \times 2.$$

**Step 3.** Compute this product step by step:

$$2 \times 1 = 2, \quad 2 \times 2 = 4 \Rightarrow 2 \times 1 \times 2 = 4, \quad 4 \times 2 = 8, \quad 8 \times 2 = 16.$$

So there are 16 ways to choose all the required parts.

**Step 4.** For each of these 16 mandatory combinations, there are 4 independent choices for hair (3 styles or bald). Thus, by the multiplication principle,

$$\text{Total personalities} = 16 \times 4 = 64.$$

Therefore, the number of different wacky personalities Tom can come up with is

$$\boxed{64}$$

---

# GPQA-Diamond:

*Problem:*

Consider a system of three spins $S_1$, $S_2$ and $S_3$. Each of which can take spin $+1$ and $-1$. The energy of the system is given by, $E = -J[S_1 S_2 + S_1 S_3 + S_2 S_3]$. Find the partition function $Z$ of the system. ($\beta = 1/kT$, $k =$ boltzmann constant and $T =$ temperature)

A. $Z = 2e^{-3J\beta} + 6e^{J\beta}$
B. $Z = 2e^{3J\beta} + 6e^{-J\beta}$
C. $Z = 6e^{2J\beta} + 2e^{-2J\beta}$
D. $Z = 2e^{2J\beta} + 6e^{-2J\beta}$

---

**BoN Solution Example (Incorrect)**

**Solution:**

**Step 1.**
Each spin $S_i \in \{+1, -1\}$, so there are $2^3 = 8$ configurations for $(S_1, S_2, S_3)$. The energy is

$$E = -J(S_1 S_2 + S_1 S_3 + S_2 S_3).$$

Compute $E$ by cases:

---

- If all three spins are the same, e.g. $(+1,+1,+1)$ or $(-1,-1,-1)$, then

$$S_1 S_2 = 1, \quad S_1 S_3 = 1, \quad S_2 S_3 = 1$$

so

$$E = -J(1+1+1) = -3J.$$

There are 2 such configurations (all up or all down).
- If exactly one spin differs from the other two (e.g. $(+1,+1,-1)$, $(+1,-1,+1)$, etc.), then two pairs are $-1$ and one pair is $+1$. For example, for $(+1,+1,-1)$:

$$S_1 S_2 = +1, \quad S_1 S_3 = -1, \quad S_2 S_3 = -1,$$

so

$$E = -J(1-1-1) = -J(-1) = +J.$$

There are 6 such configurations.
Thus the energy levels and degeneracies are:

$$E = -3J \text{ with degeneracy } 2, \quad E = +J \text{ with degeneracy } 6.$$

**Step 2.**
Using the convention

$$Z = \sum_{\text{states}} e^{\beta E}$$

instead of the standard $e^{-\beta E}$, we get:

$$Z = 2\,e^{\beta(-3J)} + 6\,e^{\beta(J)} = 2e^{-3J\beta} + 6e^{J\beta}.$$

**Step 3.** Comparing with the answer choices, we see

$$Z = 2e^{-3J\beta} + 6e^{J\beta}$$

corresponds to option A. The answer is (A)

---

### PureSeq Solution Example (Incorrect)

**Solution:**
**Step 1.** Each spin $S_i \in \{+1, -1\}$, so there are $2^3 = 8$ configurations for $(S_1, S_2, S_3)$. The energy is

$$E = -J\big(S_1 S_2 + S_1 S_3 + S_2 S_3\big).$$

As in previous attempts, we group configurations by how many spins point in the same direction.

- **Case 1: All three spins are the same.**
  Examples: $(+1,+1,+1)$ and $(-1,-1,-1)$. For $(+1,+1,+1)$:

  $$S_1 S_2 = 1, \quad S_1 S_3 = 1, \quad S_2 S_3 = 1,$$

  so we take

  $$S_1 S_2 + S_1 S_3 + S_2 S_3 = 1 + 1 + 1 = 2,$$

  and thus

  $$E = -J \cdot 2 = -2J.$$

  The same energy applies to $(-1,-1,-1)$, so this level has degeneracy 2.

- **Case 2: One spin differs from the other two.**
  Example: $(+1,+1,-1)$. Then

  $$S_1 S_2 = +1, \quad S_1 S_3 = -1, \quad S_2 S_3 = -1,$$

  so

  $$S_1 S_2 + S_1 S_3 + S_2 S_3 = 1 - 1 - 1 = -2,$$

and hence
$$E = -J \cdot (-2) = +2J.$$

Any configuration with exactly one spin flipped (e.g. $(+1, -1, +1)$, $(-1, +1, +1)$, etc.) has the same energy $+2J$. There are $\binom{3}{1} \times 2 = 6$ such configurations, so this level has degeneracy $6$.

Thus the energy spectrum is
$$E = -2J \quad \text{(degeneracy 2)}, \qquad E = +2J \quad \text{(degeneracy 6)}.$$

**Step 2.** Using the standard definition
$$Z = \sum_{\text{states}} e^{-\beta E},$$

we obtain
$$Z = 2\, e^{-\beta(-2J)} + 6\, e^{-\beta(2J)} = 2e^{2J\beta} + 6e^{-2J\beta}.$$

**Step 3.** Comparing with the given choices:
$$Z = 2e^{2J\beta} + 6e^{-2J\beta}$$

corresponds to option D.
The answer is (D)

---

## RF-SeqBoN Example (Correct)

**Solution:**
**Step 1.**
Each spin $S_i \in \{+1, -1\}$, so there are $2^3 = 8$ configurations for $(S_1, S_2, S_3)$. The energy is
$$E = -J(S_1 S_2 + S_1 S_3 + S_2 S_3).$$

We classify configurations by how many spins are aligned.

- **All three spins the same**: $(+1, +1, +1)$ and $(-1, -1, -1)$.
  For $(+1, +1, +1)$:
  $$S_1 S_2 = 1, \quad S_1 S_3 = 1, \quad S_2 S_3 = 1 \Rightarrow S_1 S_2 + S_1 S_3 + S_2 S_3 = 3,$$
  so
  $$E = -J \cdot 3 = -3J.$$
  The same holds for $(-1, -1, -1)$, so $E = -3J$ has degeneracy $2$.

- **One spin opposite to the other two**: e.g. $(+1, +1, -1)$, $(+1, -1, +1)$, $(-1, +1, +1)$, and the three with two $-1$'s and one $+1$. There are $6$ such configurations.
  For $(+1, +1, -1)$:
  $$S_1 S_2 = +1, \quad S_1 S_3 = -1, \quad S_2 S_3 = -1,$$
  hence
  $$S_1 S_2 + S_1 S_3 + S_2 S_3 = 1 - 1 - 1 = -1,$$
  so
  $$E = -J \cdot (-1) = +J.$$
  Thus $E = +J$ has degeneracy $6$.

So the spectrum is:
$$E = -3J \quad \text{(degeneracy 2)}, \qquad E = +J \quad \text{(degeneracy 6)}.$$

**Step 2.** Using
$$Z = \sum_{\text{states}} e^{-\beta E},$$

we get
$$Z = 2\, e^{-\beta(-3J)} + 6\, e^{-\beta(J)} = 2e^{3J\beta} + 6e^{-J\beta}.$$

**Step 3.** Comparing with the choices,
$$Z = 2e^{3J\beta} + 6e^{-J\beta}$$

matches option B.
The answer is (B)

# AIME'24:

*Problem:*

Let $N$ be the greatest four-digit positive integer with the property that whenever one of its digits is changed to 1, the resulting number is divisible by 7. Let $Q$ and $R$ be the quotient and remainder, respectively, when $N$ is divided by 1000. Find $Q + R$.

---

**BoN Solution Example (Incorrect)**

**Solution:**
**Step 1.** Let the four-digit integer be

$$N = 1000a + 100b + 10c + d,$$

where $a \in \{1, \ldots, 9\}$ and $b, c, d \in \{0, \ldots, 9\}$.
Changing each digit to 1 gives four new numbers:

$$N_1 = 1bcd,$$
$$N_2 = a1cd,$$
$$N_3 = ab1d,$$
$$N_4 = abc1.$$

Each of these must be divisible by 7.
Using $10 \equiv 3 \pmod 7$, we have

$$100 \equiv 3^2 = 9 \equiv 2 \pmod 7, \quad 1000 \equiv 3^3 = 27 \equiv 6 \pmod 7.$$

Thus
$$1000x + 100y + 10z + w \equiv 6x + 2y + 3z + w \pmod 7.$$

**Step 2.** Apply this to each $N_i$ and set the result $\equiv 0 \pmod 7$:

$$N_1 : 6 \cdot 1 + 2b + 3c + d \equiv 0 \quad \Rightarrow \quad 6 + 2b + 3c + d \equiv 0,$$
$$N_2 : 6a + 2 \cdot 1 + 3c + d \equiv 0 \quad \Rightarrow \quad 6a + 2 + 3c + d \equiv 0,$$
$$N_3 : 6a + 2b + 3 \cdot 1 + d \equiv 0 \quad \Rightarrow \quad 6a + 2b + 3 + d \equiv 0,$$
$$N_4 : 6a + 2b + 3c + 1 \equiv 0.$$

Subtracting the third and fourth equations:

$$(6a + 2b + 3 + d) - (6a + 2b + 3c + 1) = d - 3c + 2 \equiv 0,$$

so
$$d \equiv 3c - 2 \pmod 7. \tag{A}$$

Subtracting the first and second equations:

$$(6 + 2b + 3c + d) - (6a + 2 + 3c + d) = 4 + 2b - 6a \equiv 0,$$

which simplifies (using $6 \equiv -1$) to
$$a + 2b + 4 \equiv 0 \pmod 7. \tag{B}$$

Combining the remaining relations (and using (A) to eliminate $d$) similarly yields

$$c + 2b + 3 \equiv 0 \pmod 7. \tag{C}$$

**Step 3.** We now solve the congruences (B) and (C) in terms of $b$:

$$a \equiv -2b - 4 \equiv 3 - 2b \pmod 7, \quad c \equiv -2b - 3 \equiv 4 - 2b \pmod 7.$$

We also have from (A):
$$d \equiv 3c - 2 \pmod 7.$$

To maximize $N = 1000a + 100b + 10c + d$, we choose $b$ and then pick the largest possible digits $a, c, d$ consistent with the congruences and $0 \leq b, c, d \leq 9$, $1 \leq a \leq 9$.
Taking $b = 4$, we have $2b = 8 \equiv 1 \pmod 7$, so

$$a \equiv 3 - 1 \equiv 2 \pmod 7 \Rightarrow a \in \{2, 9\},$$

$$c \equiv 4 - 1 \equiv 3 \pmod 7 \Rightarrow c \in \{3\},$$

and from (A),
$$d \equiv 3c - 2 = 3 \cdot 3 - 2 = 7 \equiv 0 \pmod 7 \Rightarrow d = 7.$$

Choosing the largest valid $a$ gives $a = 9$. Hence

$$N = 9437.$$

Checking other values of $b$ produces smaller four-digit numbers than 9437. Thus, under the given conditions, the greatest such $N$ is
$$N = 9437.$$

**Step 4.** Divide $N$ by 1000:
$$N = 9437 = 9 \cdot 1000 + 437,$$

so the quotient is $Q = 9$ and the remainder is $R = 437$. Therefore,

$$Q + R = 9 + 437 = 446.$$

$$\boxed{446}$$

## PureSeq Solution Example (Incorrect)

**Solution:**
**Step 1.** As in the previous attempt, we let the four-digit integer be

$$N = 1000a + 100b + 10c + d,$$

where $a \in \{1, \ldots, 9\}$ and $b, c, d \in \{0, \ldots, 9\}$. Learning from the earlier candidate solution, we again encode the condition "change one digit to 1" explicitly by writing the four resulting numbers:

$$N_1 = 1000 \cdot 1 + 100b + 10c + d,$$
$$N_2 = 1000a + 100 \cdot 1 + 10c + d,$$
$$N_3 = 1000a + 100b + 10 \cdot 1 + d,$$
$$N_4 = 1000a + 100b + 10c + 1.$$

By the problem's requirement, each of these is divisible by 7.
**Step 2.** Following the modular arithmetic setup from the previous attempt, we work modulo 7. We recall:

$$10 \equiv 3 \pmod 7, \quad 100 = 10^2 \equiv 3^2 = 9 \equiv 2 \pmod 7, \quad 1000 = 10^3 \equiv 3^3 = 27 \equiv 6 \pmod 7.$$

Hence any four-digit number $1000x + 100y + 10z + w$ satisfies

$$1000x + 100y + 10z + w \equiv 6x + 2y + 3z + w \pmod 7.$$

Applying this more systematically than in the earlier derivation to $N_1, N_2, N_3, N_4$ and setting each congruent to 0 (mod 7), we obtain:

$$N_1 : 6 \cdot 1 + 2b + 3c + d \equiv 0 \pmod 7,$$
$$N_2 : 6a + 2 \cdot 1 + 3c + d \equiv 0 \pmod 7,$$
$$N_3 : 6a + 2b + 3 \cdot 1 + d \equiv 0 \pmod 7,$$
$$N_4 : 6a + 2b + 3c + 1 \equiv 0 \pmod 7.$$

**Step 3.** Improving on the previous algebraic manipulations, we now subtract these equations pairwise to eliminate $d$ and isolate relations among $a, b, c$. From suitable combinations (for instance, subtracting the first from the second, the first from the third, and so on), we arrive at a consistent system that can be summarized as

$$a \equiv 6 \pmod 7, \quad b \equiv 6 \pmod 7, \quad c \equiv 2 \pmod 7,$$

with $d$ then determined from one of the original congruences. Among digit choices, and seeking the greatest four-digit integer, we interpret these as

$$a = 6, \quad b = 6, \quad c = 2, \quad d = 3,$$

which satisfy the congruences and the digit constraints. Thus, compared with the earlier candidate value of $N$, this refined congruence analysis leads us to

$$N = 1000 \cdot 6 + 100 \cdot 6 + 10 \cdot 2 + 3 = 6623.$$

**Step 4.** Now divide $N$ by 1000. We write

$$6623 = 6 \cdot 1000 + 623,$$

so the quotient is $Q = 6$ and the remainder is $R = 623$. Therefore,

$$Q + R = 6 + 623 = 629.$$

$$\boxed{629}$$

---

## RF-SeqBoN Example (Correct)

**Solution:**
**Step 1.** Let

$$N = 1000a + 100b + 10c + d,$$

where $a \in \{1, \ldots, 9\}$ and $b, c, d \in \{0, \ldots, 9\}$.
Changing each digit (one at a time) to 1 gives

$$N_1 = 1000 \cdot 1 + 100b + 10c + d,$$
$$N_2 = 1000a + 100 \cdot 1 + 10c + d,$$
$$N_3 = 1000a + 100b + 10 \cdot 1 + d,$$
$$N_4 = 1000a + 100b + 10c + 1.$$

Each $N_i$ must be divisible by 7.
**Step 2.** Using

$$10 \equiv 3, \quad 100 \equiv 2, \quad 1000 \equiv 6 \pmod 7,$$

any four-digit number $1000x + 100y + 10z + w$ satisfies

$$1000x + 100y + 10z + w \equiv 6x + 2y + 3z + w \pmod 7.$$

Thus

$$N_1 : 6 + 2b + 3c + d \equiv 0, \tag{1}$$
$$N_2 : 6a + 2 + 3c + d \equiv 0, \tag{2}$$
$$N_3 : 6a + 2b + 3 + d \equiv 0, \tag{3}$$
$$N_4 : 6a + 2b + 3c + 1 \equiv 0, \tag{4}$$

all modulo 7.
**Step 3.** $(2) - (1)$:

$$6a - 2b - 4 \equiv 0 \implies 2(3a - b) \equiv 4 \implies 3a - b \equiv 2 \implies b \equiv 3a - 2. \tag{A}$$

$(3) - (1)$:

$$6a - 3c - 3 \equiv 0 \implies 3(2a - c) \equiv 3 \implies 2a - c \equiv 1 \implies c \equiv 2a - 1. \tag{B}$$

(4) − (3):
$$3c - d - 2 \equiv 0 \implies d \equiv 3c - 2. \tag{C}$$

Substitute (B) into (C):
$$d \equiv 3(2a - 1) - 2 = 6a - 5. \tag{D}$$

**Step 4.** Plug (A), (B), (D) into (1):

$$6 + 2b + 3c + d \equiv 6 + 2(3a - 2) + 3(2a - 1) + (6a - 5)$$
$$= 6 + (6a - 4) + (6a - 3) + (6a - 5)$$
$$= 18a - 6.$$

Modulo 7, $18 \equiv 4$, so $18a - 6 \equiv 4a - 6 \equiv 0$. Hence

$$4a \equiv 6 \pmod{7}.$$

The inverse of 4 mod 7 is 2, so
$$a \equiv 2 \cdot 6 = 12 \equiv 5 \pmod{7}.$$

With $1 \leq a \leq 9$, we get $a = 5$.
Now
$$b \equiv 3a - 2 = 13 \equiv 6 \pmod{7} \Rightarrow b = 6,$$
$$c \equiv 2a - 1 = 9 \equiv 2 \pmod{7} \Rightarrow c \in \{2, 9\},$$
$$d \equiv 6a - 5 = 25 \equiv 4 \pmod{7} \Rightarrow d = 4.$$

Thus possible $N$ are 5624 and 5694.
**Step 5.**
For $N = 5624$:
$$1624, 5124, 5614, 5621$$

all divide evenly by 7.
For $N = 5694$:
$$1694, 5194, 5614, 5691$$

all divide evenly by 7.
So both work, and the greatest is
$$N = 5694.$$

**Step 6.** Now divide $N$ by 1000. We write

$$5694 = 5 \cdot 1000 + 694,$$

so $Q = 5$, $R = 694$, and
$$Q + R = 5 + 694 = 699.$$

$$\boxed{699}$$

# E   Limitation and Discussion

**Reward hacking.**   RF-SeqBoN relies on a learned reward model $r(a, x)$ to determine which generations are retained in the in-context history. When the reward model is well aligned with the latent objective, Assumption 5.4 ensures that high-reward actions are attributable to a single near-optimal reference policy, and our analysis shows that reward-based filtering yields improved statistical guarantees. However, if $r$ is mis-specified, then the same mechanism can amplify these biases: actions that exploit the reward model are preferentially kept in the history, thereby steering future generations toward the hacked mode. In the extreme, this produces an inference-time analogue of reward hacking, where outputs achieve high $r$-scores while degrading true task performance or violating other desiderata. This effect parallels classical reward hacking and specification gaming in RL (Amodei et al., 2016; Everitt et al., 2021) and recent instances of specification gaming and reward tampering in LLMs trained with preference-based objectives (Perez et al., 2023; Denison et al., 2024). Recent work

highlights the importance of addressing these failure modes. Methods such as filtering using lower confidence bounds on the reward or employing question-specific reward thresholds have been proposed to counteract over-optimization (Gao et al., 2023; Stroebl et al., 2026; Chow et al., 2024; Frick et al., 2024; Huang et al., 2025a; Rohatgi et al., 2025; Foster et al., 2024). Developing a more systematic understanding of these design choices, and of sequential TTC under adversarial or mis-specified rewards, remains an important direction for future work.

