# OpenReview forum: "On the Limits of Test-Time Compute: Sequential Reward Filtering for Better Inference"
_ICML.cc/2026/Conference — ICML 2026 regular_

### Official Review · Reviewer_PzX5 · 2026-02-28

**Soundness:** 2
**Presentation:** 3
**Significance:** 2
**Originality:** 2
**Overall Recommendation:** 3
**Confidence:** 3

**Summary:**

The paper investigates test-time compute (TTC) strategies for large language models, comparing parallel Best-of-N (BoN) sampling with sequential approaches. The authors attempt to address a major challenge in determining how to optimally allocate inference-time computation to improve model outputs. Specifically, the authors attempt to address the question regarding the theoretical limitations of parallel BoN under a fixed budget. The paper introduces a theoretical framework based on a mixture-of-reference-policies model to establish sample complexity lower bounds for parallel TTC. Based on this, it proposes Reward-Filtered Sequential Best-of-n (RF-SeqBoN), a procedure that sequentially generates answers, evaluates them with an external reward model, and appends generations scoring above a specific threshold back into the LLM's context window. The theoretical claims are evaluated empirically across math and reasoning benchmarks.

**Compliance With Llm Reviewing Policy:**

Affirmed.

**Key Questions For Authors:**

Please refer to Strengths And Weaknesses.

**Limitations:**

Yes.

**Strengths And Weaknesses:**

**Strengths:**
* The paper addresses the allocation and scaling of test-time compute, which is a meaningful and relevant problem in current LLM research.
* The theoretical framing of sequential test-time compute through a mixture-of-reference-policies model provides an interesting analytical perspective on in-context policy shifts.


**Weaknesses:**
* The theoretical framework relies heavily on an unrealistic assumption regarding the pretraining data distribution (Remark 4.1). The paper assumes that pretraining trajectories consist of an initial prompt followed by multiple independent answers drawn from the same reference policy, but real-world LLM pretraining corpora never contain this specific structure in practice. The authors need to logically or empirically justify this assumption.
* There is a disconnect between the theoretical formulation and the empirical implementation. The mathematical proofs model the performance gain as a probabilistic shift caused by purely appending high-reward trajectories (Section 5). However, the actual experiments (Appendix D.2 and D.3.1) utilize explicit natural language meta-instructions instructing the model to self-correct based on previous attempts. It is unclear if the empirical gains are driven by the proposed latent Bayesian mechanism or simply by standard instruction-following capabilities. The authors should clarify if the method still improves without these explicit meta-instructions.
* The evaluation of computation time and budget efficiency (Appendix D.4, Figure 5) does not fully reflect the latency bottleneck of single queries. RF-SeqBoN requires sequential generation, sequential PRM scoring, and computing the KV-cache for a growing context window, whereas BoN can be executed in parallel. To better support the efficiency claims, the authors should provide wall-clock latency comparisons for single queries and discuss the computational overhead of the growing KV-cache.
* The method relies on a fixed, dataset-specific reward threshold ($\gamma$). Table 2 shows that $\gamma$ requires careful tuning per dataset (ranging from 0.90 to 0.97). The paper lacks a discussion on how to dynamically determine an appropriate $\gamma$ in real-world deployment scenarios where the domain and difficulty of queries are unknown in advance.

---

> ### Author Rebuttal · Authors · 2026-03-31
>
> Thank you very much for your thoughtful review and valuable suggestions. Below, we briefly summarize your main concerns and our responses.
>
> ---
>
> **Q1**:
> The pretraining-data assumption is too idealized and does not match real LLM pretraining corpora.
>
> **A1**:
> We would like to clarify that the pretraining assumption in *Remark 4.1* is not meant as a literal description of real-world pretraining corpora, but as a conceptual abstraction that captures an important structural property of LLMs.
>
> Specifically, assuming each trajectory is generated from a single reference policy formalizes the idea that LLMs implicitly encode a mixture of latent generation behaviors, such as different reasoning styles, verbosity levels, or solution strategies. Although real-world corpora are not explicitly organized this way, LLMs trained on diverse data sources often exhibit heterogeneous generation patterns that can be naturally interpreted as arising from a mixture of latent behaviors. Under this view, our assumption provides a tractable way to model the conditional distribution $\pi_{\mathrm{LLM}}(\cdot \mid h)$ as a posterior mixture over latent policies, where conditioning on history shifts probability mass toward more consistent components.
>
> This perspective is also consistent with large-scale pretraining datasets such as *The Pile* and *FLAN*, which can be viewed as mixtures of diverse sources, styles, or task distributions. We will revise the paper to make this motivation more explicit and to emphasize that the assumption is a conceptual approximation rather than a literal data-generation model.
>
> ---
>
> **Q2**:
>
> The theory attributes the gain to appending high-reward histories, but the experiments also use explicit self-correction instructions. Does the method still help without such meta-instructions?
>
> **A2**:
> To address it directly, we ran an additional ablation in which we replaced the original self-correction prompt (line 1406) with the neutral sentence:
>
> > “The previous candidate solutions above are provided only as extra context reference.”
>
> Under this updated prompt, the model is simply given prior accepted solutions as context and asked to solve the problem from scratch, without any explicit self-correction-style meta-instruction. The corresponding result on MATH500 with the Qwen3-4B-Instruct backbone is provided in the link below:
> > <https://anonymous.4open.science/r/F27K81/fig2_prompt_ablation.pdf>
>
> The dashed curves (updated prompt) closely match the solid curves (original prompt), and RF-SeqBoN still outperforms both BoN and PureSeq as the generation budget increases. This suggests that the gain is not primarily driven by prompt-engineering or explicit self-correction instructions, but is consistent with the core mechanism of our method: conditioning on selectively retained high-reward histories. We will clarify this distinction more explicitly in the revision.
>
> ---
>
> **Q3**:
>
> The computation-time analysis does not reflect real wall-clock latency for a single query, since RF-SeqBoN has sequential generation, sequential PRM scoring, and KV-cache overhead, while BoN can be parallelized.
>
> **A3**:
> We agree that *Figure 5* of the *Appendix D.4* does not fully capture the single-query latency tradeoff. In our implementation, however, the sequential methods use a small capped history window throughout: we set $\texttt{history\\_budget} = 3$ (*Table 1*, *Appendix D.1*), so at most three previous solutions are kept in context. Thus, the KV-cache/context overhead is present but explicitly bounded.
>
> To address your concern directly, we added a wall-clock latency vs. accuracy analysis. Specifically, we measure the average first-token latency for a single query at generation budget $N=128$, using a single Nvidia A6000 GPU and the Qwen3-4B-Instruct backbone, and pair it with accuracy on MATH500 Levels 3–5. The resulting Pareto-style figure is provided here:
> > <https://anonymous.4open.science/r/F27K81/fig3_latency_pareto.pdf>
>
> The figure shows that BoN has the lowest latency, as expected, but RF-SeqBoN achieves a better latency–accuracy tradeoff than PureSeq, and on the harder Level 4–5 subsets it attains noticeably higher accuracy than BoN at a moderate latency cost. Our claim is therefore not that RF-SeqBoN minimizes wall-clock latency, but that it offers a favorable Pareto tradeoff under bounded context-growth overhead.
>
> ---
>
> **Q4**:
> The method depends on dataset-specific tuning of $\gamma$, and the paper does not explain how to choose it when the test domain is unknown.
>
> **A4**:
>
> Although the best-performing $\\gamma$ varies by dataset (Table 2), the range is narrow ($0.90$--$0.97$). We also tested a fixed $\gamma=0.94$ across all five benchmarks:
> > <https://anonymous.4open.science/r/F27K81/fig4_fixed_gamma.pdf>
>
> The fixed threshold remains consistently competitive, suggesting limited sensitivity to the exact choice of $\\gamma$. Please see **A2 for Reviewer fAEK** for additional details, due to the character limit here.

---

> > ### Author Rebuttal · Reviewer_PzX5 · 2026-04-01
> >
> > I would like to thank the authors for the rebuttal, which addresses most of my experimental concerns. However, two substantive issues remain. First, the mixture-of-reference-policies assumption remains a strong idealization whose practical validity for modern LLMs is unclear. Second, the core algorithmic contribution — appending high-reward responses with a fixed threshold — is relatively straightforward, and I am not fully convinced by the technical novelty. I am raising my score from 2 to 3.

---

### Official Review · Reviewer_H85w · 2026-03-06

**Soundness:** 3
**Presentation:** 3
**Significance:** 3
**Originality:** 2
**Overall Recommendation:** 5
**Confidence:** 3

**Summary:**

The paper presents the Reward-Filtered Sequential Best-of-n (RF-SeqBoN) framework for TTC under a mixture-of-policies model. The core idea of RF-SeqBoN lies in generating outputs sequentially and feeding only high-reward results back into the model’s context, thereby progressively shifting the model’s output distribution toward the reward-optimal policy. By providing both theoretical and practical analysis, the authors emphasize that the proposed method derives stronger theoretical guarantees and practical accuracy than standard TTC paradigms like BoN or PureSeq.

**Compliance With Llm Reviewing Policy:**

Affirmed.

**Key Questions For Authors:**

1 - Can the authors provide the value of \epsilon used in the experiment to analyze the lower bound on the test-time budget and the lower bound value for BoN, RF-SeqBoN, with the corresponding \epsilon? Are these lower bounds relevant to the experiment results in Figure 1?

2 – Can the author provide a practical example to verify the claims in Theorem 5.1 and Remark 5.2?

**Limitations:**

yes

**Strengths And Weaknesses:**

Strengths: Overall, the paper is clearly written, well-structured, and technically solid. The topic is timely and relevant for ICML, considering the increasing interest in using test-time compute to enhance the performance of large language models. The validation is dense and thorough, showing the practical efficiency of the proposed method in several setups.

Weaknesses: Although several experiments were conducted on the empirical side, the analysis of theoretical claim on the lower bound on the test-time budget required to achieve a near-optimal policy has not been clearly explain.

---

> ### Author Rebuttal · Authors · 2026-03-31
>
> We sincerely thank you for your careful reading and helpful feedback. Below, we summarize your key questions and concerns and provide our corresponding responses.
>
> ---
>
> **Q1**:
> What $\epsilon$ is used in the lower-bound analysis, what lower-bound values it gives for BoN and RF-SeqBoN, and whether those theoretical lower bounds are actually reflected in the empirical trends shown in *Figure 1*.
>
> **A1**：
> Thank you for raising this important question. We would like to clarify that the lower bounds derived in our paper are information-theoretic guarantees, rather than quantities that can be directly instantiated or measured in experiments.
>
> In the lower bound proof, $\epsilon$ represents the target accuracy level in terms of regret, and the lower bound establishes the existence of hard instances where any parallel TTC method may require strictly more samples than sequential TTC to achieve $\epsilon$-optimality. Therefore, the result should be interpreted as a worst-case sample complexity separation, rather than a quantity that can be directly computed for a given task or model.
>
>
>
> In our experiments, our experiments evaluate performance using task-specific metrics such as exact-match. We do not assume that real-world tasks correspond to the hard instances constructed in the lower bound. Rather, the role of the theory is to explain the potential advantage of sequential methods, while Figure 1 provides empirical evidence that is consistent with this prediction, showing improved performance of our method under the same test-time budget.
>
> ---
>
> **Q2**:
> A practical example to verify the claims in *Theorem 5.1* and *Remark 5.2*.
>
> **A2**: *Theorem 5.1* and *Remark 5.2* aim to demonstrate a strict advantage over the parallel TTC guarantee of Huang et al., under the mixture-of-reference-policies model and the sequential sample-and-evaluate framework. To further support this claim, we provide the following constructive example.
>
> Consider a single prompt $x$ and a binary action space $A = \\{ a^\star, a^{\prime} \\}$, where $r(a^\star, x)=1$ and $r(a^{\prime}, x)=0$. Suppose there are two reference policies $\pi_1$ and $\pi_2$ such that $\pi_1(a^\star)=p \gg q=\pi_2(a^\star)$.
>
> Under our model, the LLM policy is a history-dependent mixture satisfying
> $$
> \pi_{\mathrm{LLM}}(a^\star \mid x)
> = p \cdot P(\tau=1 \mid x) + q \cdot P(\tau=2 \mid x).
> $$
>
> Now consider a history
> $$
> h = \bigl(x, \underbrace{a^\star, a^\star, \ldots, a^\star}_{k\ \text{times}}\bigr).
> $$
>
> By Bayes' rule (lines 255--262), we have
> $$
> P(\tau=i \mid h) \propto \pi_i(a^\star \mid x)^k\cdot p_{\mathrm{ref}}(i).
> $$
>
> Since $p \gg q$, it follows that $P(\tau=1 \mid h) \to 1$ as $k \to \infty$. Therefore,
> $$
> \pi_{\mathrm{LLM}}(a^\star \mid h) \to p \ge \pi_{\mathrm{LLM}}(a^\star \mid x).
> $$
>
> Thus, when $\pi^\star = \mathbf{1}(a^\star \mid x)$ is the optimal policy, we have
> $$
> M^\epsilon_{\pi^\star(\cdot \mid x), \pi_{\mathrm{LLM}}(\cdot \mid h)}
> \le
> M^\epsilon_{\pi^\star(\cdot \mid x), \pi_{\mathrm{LLM}}(\cdot \mid x)},
> $$
> since it captures coverage, and $\pi_{\mathrm{LLM}}(a^\star \mid h)$ is much larger.
>
> Substituting this inequality into *Theorem 5.1* shows that conditioning on informative histories leads to a strictly improved bound compared to the parallel TTC setting. This provides a concrete illustration of the improvement stated in *Remark 5.2*.

---

> > ### Author Rebuttal · Reviewer_H85w · 2026-03-31
> >
> > The answers are ok for me. I let my score as it is.

---

### Official Review · Reviewer_fAEK · 2026-03-09

**Soundness:** 3
**Presentation:** 3
**Significance:** 3
**Originality:** 3
**Overall Recommendation:** 4
**Confidence:** 4

**Summary:**

This paper studies on test-time scaling (TTC), and demonstrates that Best-of-N is suboptimal through a mixture-of-reference policy. The authors propose a method called Reward-Filtered Sequential Best-of-n (RF-SeqBoN), which only feeds high-reward generations back into the context. The paper includes both theoretical and empirical analysis. On the theory side, it separates parallel and sequential TTC under its assumptions; on the empirical side, it evaluates RF-SeqBoN against BoN and a pure sequential baseline on five verifiable reasoning benchmarks using three backbone models, and demonstrate the effectiveness of RF-SeqBoN.

**Compliance With Llm Reviewing Policy:**

Affirmed.

**Final Justification:**

The rebuttal has resolved my concerns.

**Key Questions For Authors:**

Please refer to the weaknesses above.

**Limitations:**

yes.

**Strengths And Weaknesses:**

Strengths:

- The paper studies TTC from an interesting perspective. It studies whether parallel compute is intrinsically worse than sequential compute when guided by a reward model.
- The author's proposed method is simple and clear, with only operations like sample, score, and append if the reward passes a certain threshold.
- The authors conduct theoretical analysis on why history-conditioned generation could outperform BoN with their mixture-of-reference-policies formalization, which makes the paper more technically sound.
- Empirical results demonstrate that their proposed method can be well-generalized to across three backbone models on several reasoning benchmarks. The authors also include comprehensive analysis and ablations on the reward threshold, a weaker ORM reward model, matched-computation-time comparison and so on.
- The paper's presentation is clear and generally easy to follow.

Weakness:
- My main concern is that the authors mainly compared with BoN and PureSeq and TopK. These are valid baselines, but are only variants of their proposed work. As the authors have cited so many related works (sequential and reward-based ones), I hope the authors can compare with these stronger baselines under a matched token budget and reward model setting.
- Does the optimal choice of gamma vary on different datasets?

---

> ### Author Rebuttal · Authors · 2026-03-31
>
> Thank you very much for your thoughtful comments and constructive suggestions. Below, we provide a concise summary of your main questions and concerns, followed by our responses.
>
> ---
>
> **Q1**:
> Comparison with stronger baseline algorithms (cited sequential and reward-based works), under a matched token budget and reward model setting.
>
> **A1**:
> Thank you for this helpful suggestion. We added comparisons with three stronger baselines under a matched token budget: Beam Search, DVTS, and PAG. *Beam Search* is a verifier-guided tree-search method. Following Snell et al. [1], it uses a process reward model (PRM) to score intermediate reasoning steps during search, and then applies final answer selection over the completed candidates. *DVTS (Diverse Verifier Tree Search)* is a stronger tree-search variant from the Hugging Face implementation/blog [2]: it extends beam search by splitting the initial beams into multiple independent subtrees and expanding them greedily with PRM scores, which improves diversity and tends to help at larger test-time budgets. *PAG (Policy as Generative Verifier)* [3] is a strong sequential self-correction baseline: a single model alternates between policy and verifier roles and follows a selective verify-then-revise procedure, revising only when its own generative verification step detects an error.
>
> We compare these three methods against RF-SeqBoN using the same Qwen3-4B-Instruct backbone on MATH500. The additional result is provided in the anonymous figure-only link below:
> > <https://anonymous.4open.science/r/F27K81/fig1_stronger_baselines.pdf>
>
> As shown in the figure, RF-SeqBoN remains highly competitive against all three stronger baselines across the full range of generation budgets, while using a substantially simpler mechanism than the tree-based search methods. In particular, Beam Search and DVTS rely on PRM-guided branching over multiple search paths, whereas RF-SeqBoN uses a much simpler single-sequence reward-filtering mechanism. We will add this stronger-baseline comparison and corresponding discussion in the revision.
>
> **References:**
> > [1] Snell, C., Lee, J., Xu, K. and Kumar, A., 2024. Scaling llm test-time compute optimally can be more effective than scaling model parameters. arXiv preprint arXiv:2408.03314.
> >
> > [2] Beeching, E., Tunstall, L., and Rush, S. Scaling Test-Time Compute with Open Models. Hugging Face blog / Search-and-Learn repository, 2024.
> >
> > [3] Jiang, Y., Xiong, Y., Yuan, Y., Xin, C., Xu, W., Yue, Y., Zhao, Q. and Yan, L., 2025. Pag: Multi-turn reinforced llm self-correction with policy as generative verifier. arXiv preprint arXiv:2506.10406.
>
> ---
>
> **Q2**:
> Does the optimal choice of gamma vary on different datasets.
>
> **A2**:
> Yes. As reported in *Table 2* in *Appendix D.1*, the best-performing choice of $\gamma$ does vary across benchmark datasets. That said, the variation is fairly limited: the optimal values all fall in a relatively narrow range, approximately $0.90$ to $0.97$. This suggests that RF-SeqBoN is not overly sensitive to dataset-specific tuning.
>
> To further address this concern, we also evaluated a fixed threshold $\gamma=0.94$ across all five benchmarks using the same Qwen3-4B-Instruct backbone. The additional result is provided in the anonymous figure-only link below:
> > <https://anonymous.4open.science/r/F27K81/fig4_fixed_gamma.pdf>
>
> The figure shows that a single fixed $\gamma$ remains consistently competitive with the dataset-specific optimal choice, with only a small performance gap in most cases. This suggests that while per-dataset tuning can provide modest improvements, RF-SeqBoN is not highly sensitive to $\gamma$ and can still work well with a shared threshold in deployment settings where the task distribution is unknown.

---

> > ### Author Rebuttal · Reviewer_fAEK · 2026-04-03
> >
> > Thanks. The rebuttal has resolved my concerns.

---

### Decision · Program_Chairs · 2026-04-30

**Decision:**

Accept (regular)

**Comment:**

The paper addresses a question in test-time compute for large language models: whether sequentially allocated compute can outperform parallel Best-of-N under a fixed budget. Reviewers generally found the work well-motivated, clearly written, and technically solid. The main strengths are the interesting theoretical perspective based on a mixture-of-reference-policies model, the simplicity and clarity of the proposed method, and a fairly thorough empirical evaluation across multiple backbones and reasoning benchmarks. Reviewers also appreciated the additional ablations and clarifications provided in the rebuttal, with one reviewer explicitly stating that their concerns were resolved.

Overall, the consensus is positive.